# Emergent Corpus Pre-training Benefits Vision Language Models

**Makanjuola Ogunleye**                                    *mogunleye@vt.edu*
*Virginia Tech*

**Chase Vickery**                                          *cvickery@vt.edu*
*Virginia Tech*

**Ismini Lourentzou**                                      *lourent2@illinois.edu*
*University of Illinois Urbana-Champaign*

**Reviewed on OpenReview:** *https://openreview.net/forum?id=bivKGSaXkD*

## Abstract

Vision-Language Pre-trained Models (VL-PTMs) have achieved impressive performance across a wide range of tasks, but their success often hinges on access to large-scale multimodal datasets. While effective in high-resource settings, these models tend to struggle in data-scarce regimes. In this work, we investigate Emergent Communication (EC) as a mechanism to improve sample efficiency in VL-PTMs. We pre-train a Vision-Language Model (VLM) using EC tokens generated through a referential game between two artificial agents. Across three diverse cross-modal matching and reasoning benchmarks, EC pretraining yields substantial gains, improving Visual Referring Expression (VRE) accuracy by 108.6% and Visual Entailment (VE) by 69.6%. To further validate the effectiveness of EC pretraining, we introduce LLaVA-1.5-EC, a LLaVA variant trained entirely on EC tokens. LLaVA-1.5-EC outperforms strong LVLM baselines, including BLIP-2 (13B), achieving relative gains of 104.23% on VizWiz, 34.8% on GQA, and 10.8% on VQAv2, and top performance on MMBench, a challenging instruction-following benchmark. These results highlight the transferability and generalization capacity of EC pretraining and underscore the potential of leveraging grounded EC tokens to enhance vision-language reasoning in low-resource settings, especially in settings with limited natural language data. We discuss implications and propose avenues for future research to explore the connections between EC and VL for multimodal understanding and effective human-machine communication.

🌐 **Project Website:** https://plan-lab.github.io/ec-vlm/

## 1 Introduction

Vision-Language Pre-trained Models (VL-PTMs) have achieved remarkable progress across tasks such as image-text retrieval, referring expression comprehension, and visual question answering (Li et al., 2023a; Liu et al., 2023a). The prevailing approach has been to build larger models, incorporating more parameters in order to enhance their generalizability (Du et al., 2022). Alternative efforts have been directed toward devising improved architectures for the fusion of text and image representations, such as fusion encoders, dual encoders, or a combination thereof (Radford et al., 2021; Li et al., 2020a). Despite these advancements, robust generalization in low-resource settings remains a persistent challenge, largely due to the scarcity of aligned vision-language data across many real-world tasks (Xu et al., 2023).

Self-supervised learning (SSL) has emerged as a key strategy for addressing label scarcity in vision-language learning, with models like CLIP (Radford et al., 2021) and VisualBERT (Li et al., 2019) learning cross-modal representations via contrastive or pretext tasks. However, SSL methods encounter several challenges that

can limit practical utility, *e.g.*, task design dependency, computational complexity, sensitivity to the choice of hyper-parameters, and catastrophic forgetting (Purushwalkam et al., 2022; Tian et al., 2020; Ericsson et al., 2021). Recent advances in discrete latent representation learning, such as Vector Quantized Variational Autoencoders (VQ-VAE) (Van Den Oord et al., 2017) and unified tokenization schemes (Ge et al., 2024), have shown promise in bridging the modality gap. While VQ-VAE relies on two-stage training and gradient approximations that can hinder optimization, unified tokenizers typically introduce early modality fusion that may limit flexibility for task-specific adaptation. Both approaches also assume access to large-scale vision-language pairs or pretrained model weights, which restricts their utility in low-resource settings.

Emergent Communication (EC) offers a compelling alternative for discrete representation learning in multi-modal contexts, particularly in settings where labeled data is scarce. EC explores how artificial agents develop structured, functional communication protocols through interaction, typically in service of collaborative task completion. In multi-agent environments, agents are jointly trained to achieve a shared objective by max-imizing cumulative reward, with communication emerging as a means to improve coordination (Lazaridou et al., 2017; 2018; Havrylov & Titov, 2017). A common instantiation of EC is the referential game (Evtimova et al., 2018; Lazaridou et al., 2018; Li et al., 2020b; Yao et al., 2022), where a Speaker agent observes an image and generates a discrete message intended to describe it. The Listener agent receives this message and must identify the correct image from a set of distractors. Through repeated interaction and shared supervision, agents converge on a discrete token protocol that reflects visual structure and supports grounded inference.

In the context of vision-language pretraining and Vision Language Models (VLMs), Emergent Communica-tion (EC) offers a promising alternative to natural language supervision, particularly in resource-constrained settings. EC tokens have been shown to exhibit key linguistic properties such as compositionality, structure, and symbolic abstraction (Yao et al., 2022; Li et al., 2020b), while being learned entirely from interaction. As such, EC provides a grounded and task-driven learning signal that can substitute for or complement human-annotated text. If effective, EC could serve as a scalable strategy for pretraining vision-language models when parallel image-text data is limited or unavailable. More broadly, studying EC in this setting may shed light on how communication protocols emerge in artificial agents, offering a new lens on the inductive biases underlying vision-language understanding.

Building on prior work demonstrating the benefits of emergent communication in language modeling (Li et al., 2020b), we extend EC to vision-language pretraining, demonstrating its broader applicability in mul-timodal learning. Specifically, we examine how EC tokens, learned through interaction alone, can serve as an effective pretraining signal for downstream vision-language tasks. Through a comprehensive series of exper-iments, we evaluate the transferability of EC pretraining across diverse VL tasks, including Visual Referring Expression (VRE), Visual Entailment (VE), Visual Question Answering (VQA), and Image Captioning (IC). For example, EC pretraining improves OFA's (Wang et al., 2022a) VRE accuracy from 29.8% to 62.2%, and VE accuracy from 50.3% to 85.3%, showcasing strong task transfer from EC-generated text. These results in-dicate that EC tokens effectively capture visual context and support fine-grained semantic reasoning. To fur-ther underscore the robustness and importance of EC pretraining, we construct LLaVA-1.5-EC, a variant of LLaVA-1.5 (Liu et al., 2024a) trained entirely on EC tokens. Despite being trained without natural language captions, LLaVA-1.5-EC outperforms strong vision-language baselines across task-oriented and instruction-following benchmarks, highlighting the transferability and generalization potential of EC pretraining at scale.

These findings demonstrate the broad potential of EC pretraining for vision-language modeling, particularly in low-resource or weakly supervised settings. The contributions of our work can be summarized as follows:

**(1)** We introduce EC-VLM, a vision-language pretraining framework that employs Emergent Commu-nication (EC) between agents to generate supervision signals for pretraining. We demonstrate that EC pretraining transfers effectively to diverse downstream multimodal tasks.

**(2)** We empirically show that EC tokens encode structured and compositional semantics that generalize across tasks and modalities, positioning EC as a scalable, annotation-free alternative to natural language supervision in multimodal learning.

**(3)** We provide insights into the structure and transferability of emergent language for vision-language pretraining and outline future research opportunities at the intersection of EC, multimodal repre-sentation learning, and human-machine communication.

## 2 Related Work

### 2.1 Emergent Communication

Recently, there has been a growing interest in investigating the emergence of language within deep agent networks for task completion (Lazaridou et al., 2017; 2018; Lazaridou & Baroni, 2020; Mordatch & Abbeel, 2018; Raviv et al., 2019b; Das et al., 2019). Pioneering works simulate communication while training agents (Lazaridou et al., 2017; Evtimova et al., 2018). Building upon these foundations, Mordatch & Abbeel (2018); Raviv et al. (2019a) further explored the compositionality aspect of the emerged language. Several works have also attempted to interpret the structure and linguistic properties of the emergent language. For instance, Chaabouni et al. (2019) discovered that networks develop an anti-efficient encoding scheme, where longer messages are associated with the most frequent inputs, contrary to human language, where the most frequent words are usually represented by shorter strings. Additionally, Patel et al. (2021) observed the emergence of egocentric grounded messages when agents were initialized in complex environments. In our work, we leverage emergent language as a source of *inductive bias* and investigate its potential to enhance *cross-modal vision and language learning*. To this end, we explore the benefits of Emergent Communication (EC) for vision-language pre-training and, interestingly, we gain valuable insights that contribute to a better understanding of EC language dynamics.

### 2.2 Corpus Transfer of Synthetic or Emergent Communication Language

While significant progress has been made in analyzing the properties of emergent communication language (Mordatch & Abbeel, 2018; Resnick et al., 2020; Lazaridou & Baroni, 2020; Chaabouni et al., 2019), a different line of research has explored the potential transfer benefits of synthetic, artificial, or emergent language in improving learning and generalization in language models (Yao et al., 2022; Li et al., 2020b; Papadimitriou & Jurafsky, 2020). For instance, Papadimitriou & Jurafsky (2020) investigated the impact of language structure on model learning, finding that exposure to diverse structured data, such as music, Java code, and nested symbols, can improve transferability to natural language. Similarly, Yao et al. (2022) demonstrated the efficacy of pre-training on emergent languages for downstream natural language tasks, achieving performance comparable to models trained on natural language data, notably in low-resource settings. Furthermore, Li et al. (2020b) showed that emergent communication protocols, even when pre-trained without human language data, can benefit downstream NLP applications such as machine translation, leading to improved accuracy and efficiency. We enrich this body of work by investigating the benefits of EC pertaining to Vision-Language learning. Specifically, we investigate whether a corpus of EC tokens, generated through referential games grounded in both vision and language, can serve as an effective pretraining signal for cross-modal learning. Our results show that EC corpus transfer significantly enhances downstream performance in vision-language tasks.

### 2.3 Pre-training for Vision Language Transfer Learning

In cross-modal machine learning, the standard practice has been to pre-train vision-language models on multimodal data and then fine-tune them on downstream tasks (Wang et al., 2022a; Gan et al., 2022; Du et al., 2022) such as Visual Question Answering (Antol et al., 2015), Phrase Grounding (Yu et al., 2016), and Image Captioning (Lin et al., 2014). Several successful architectures have been proposed to model cross-modal (visual-linguistic) interactions (Li et al., 2020a; 2019). For instance, fusion encoders (Li et al., 2019; Su et al., 2020) apply attention mechanisms over joint image-text inputs, while dual encoders (Radford et al., 2021; Wang et al., 2022b) process each modality with separate encoders, making them highly efficient for tasks like image-text retrieval (Du et al., 2022; Lu et al., 2019). Different pretraining paradigms have also emerged. Contrastive learning, exemplified by CLIP (Radford et al., 2021), aligns image-text pairs in a shared embedding space using large-scale caption datasets. Generative modeling via image captioning has also proven effective. Tschannen et al. (2024) shows that pretraining vision encoders using only captioning can yield robust results across classification and vision-language benchmarks. Other common objectives include Cross-Modal Masked Language Modeling (MLM)(Liu et al., 2023b; Kim et al., 2021), where masked text tokens are predicted using visual context, and Masked Region Prediction (MRP)(Chen et al., 2020),

which involves predicting masked regions in the image from surrounding features. While much of the field has focused on scaling models, refining architectures, or optimizing pretraining objectives, this work explores fundamental questions on the potential of EC as a powerful paradigm for pre-training vision language tasks.

## 3 Method

**Lewis Referential Game.** The Lewis referential game involves two agents: a speaker S and a listener L. The speaker observes a target image sampled from the environment and sends a message intended to describe it. The listener receives this message and must identify the correct image from a set of distractors. The game's objective is for the listener to successfully recover the speaker's input, and success is measured by the listener's accuracy in selecting the correct image. This setup is a standard framework for studying emergent communication in artificial agents, following prior work (Lazaridou et al., 2018; Li et al., 2020b; Lazaridou & Baroni, 2020; Yao et al., 2022). The speaker and listener are jointly trained to maximize a shared reward, encouraging the development of an effective communication protocol through repeated interaction.

**Speaker's Message.** At each training step, an input image feature $\mathbf{I}_i \in \mathbb{R}^d$ is randomly selected from the entire set of feature representations for $N$ images $\mathcal{D} = \{\mathbf{I}_i\}_{i=1}^N$, where $d$ denotes the dimensionality of the image feature vectors. Similarly, a set of $K$ confounding images (*i.e.*, the distractors) $C_i = \{\mathbf{I}_{ij}\}_{j=1, j \neq i}^K$ is selected from $\mathcal{D}$. The speaker $\mathtt{S} : \mathbb{R}^d \to V^T$, takes the input image feature $\mathbf{I}_i$ and generates a message $\mathbf{M}_i = \langle m_1, m_2, m_i, \ldots, m_T \rangle, m_i \in V$, a sequence of discrete symbols that describes the image, where $T$ is the sequence length limit and $V$ is the message's vocabulary size. The generation process ends when either of the two conditions is satisfied: the special end-of-sentence symbol [EOS] is generated, or the maximum message length $T_{max}$ is reached. Initially, at $t = 0$, $m_0 = $ [CLS] and $h_{\mathtt{S}}^0 = \mathbf{I}_i$. At each time step $t > 0$, the generation of the $i$-th speaker message token $m_i^t$ can be described by

$$\mathbf{h}_{\mathtt{S}}^t = \mathrm{GRU}_{\mathtt{S}}\left(m_i^{t-1}, \mathbf{h}_{\mathtt{S}}^{t-1}\right), \tag{1}$$

$$m_i^t = \mathrm{Gumbel\text{-}Softmax}\left(\mathrm{MLP}_{\mathtt{S}}(\mathbf{h}_{\mathtt{S}}^t)\right), \tag{2}$$

where the Gumbel-Softmax trick (Jang et al., 2017) is employed to draw samples from categorical distributions of emergent tokens in an end-to-end differentiable way. Here, $\mathbf{h}_{\mathtt{S}}^t$ stands for the hidden state at time step $t$, while $\mathrm{MLP}_{\mathtt{S}}$ denotes the multilayer perception speaker S utilizes to project each hidden state into vectors with dimensionality equal to the vocabulary size of the emergent language.

**Listener's Inference.** The listener agent $\mathtt{L} : V^T \times \mathbb{R}^{K \times d} \to \{1, \ldots, K\}$ tries to guess the correct Image $\mathbf{I}_i$ from the set of K confounding images $C_i$, after receiving the generated speaker message $\mathbf{M}_i$. To do so, the listener utilizes a GRU layer to decode the speaker's generated message, *i.e.*,

$$\mathbf{h}_{\mathtt{L}}^0 = \mathrm{GRU}_{\mathtt{L}}\left(m_0, \mathbf{0}\right), \tag{3}$$

$$\mathbf{h}_{\mathtt{L}}^t = \mathrm{GRU}_{\mathtt{L}}\left(m_i^t, \mathbf{h}_{\mathtt{L}}^{t-1}\right), \tag{4}$$

where $\mathbf{h}_{\mathtt{L}}^t$ represents the listener's hidden state at time step $t$.

**Speaker-Listener Optimization.** The listener scores each candidate in the image set, which consists of all distractor images and the correct reference image. Given a message $\mathbf{M}$ and an image $\mathbf{I} \in \{\mathbf{I}_i \cup C_i\}$, the score is defined as

$$\mathrm{score}\left(\mathbf{M}, \mathbf{I}\right) = \left\| \mathbf{h}_{\mathtt{L}}^{|\mathbf{M}|} - \mathrm{MLP}_{\mathtt{L}}(\mathbf{I}) \right\|_2^{-2}. \tag{5}$$

The likelihood of a selected image $\mathbf{I}_j$ is:

$$p\left(\mathbf{I}_j | \mathbf{M}_i, \mathbf{I}_j, C_i\right) = \left( \frac{e^{\mathrm{score}(\mathbf{M}, \mathbf{I}_k)}}{\sum\limits_{\mathbf{I}_k \in \mathbf{I}_i \cup C_i} e^{\mathrm{score}(\mathbf{M}, \mathbf{I}_k)}} \right), \tag{6}$$

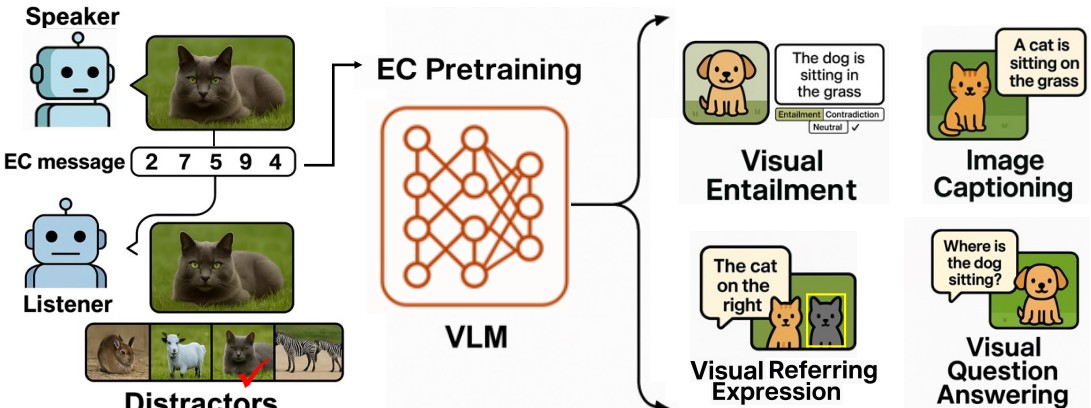

Figure 1: **Overview of our EC-VLM Pretraining Framework.** A speaker-listener pair engages in a referential game, where the speaker generates an Emergent Communication (EC) message to describe a target image, and the listener must identify the correct image among distractors. The resulting EC tokens serve as pretraining supervision for a Vision-Language Model (VLM). This EC-pretrained VLM is then fine-tuned on a range of downstream vision-language tasks, including Visual Entailment, Visual Referring Expression, Image Captioning, and Visual Question Answering. The framework enables transferable visual grounding from synthetic EC messages to natural language tasks.

where softmax sampling across scores is used to select an image from the candidate set. Speaker and listener parameters are jointly optimized by maximizing the expected log-likelihood

$$\mathcal{L} = -\mathbb{E}_{\mathbf{M}}\mathbb{E}_{\mathbf{I}_j}\left[\log p\left(\mathbf{I}_j|\mathbf{M}, \mathbf{I}_i, C_i\right)\right]. \tag{7}$$

Once training is complete, the speaker can generate a corpus of emergent language (EC) by producing discrete messages conditioned on input images. In this work, we investigate whether such a corpus can be leveraged to pretrain models for vision-language (VL) tasks and whether the structure of EC tokens offers benefits for multimodal learning. We hypothesize that EC tokens, although globally scoped, may implicitly encode information relevant to specific regions or semantic elements in the image.

As shown in Figure 1, to evaluate EC corpus transfer on cross-modal matching and reasoning downstream tasks, we generate EC messages for the images in various vision-language benchmarks: Visual Referring Expression (VRE), Visual Question Answering (VQA), and Visual Entailment (VE), and Image Captioning (IC). In VRE, the agent is presented with an image along with a referring expression that identifies a specific object or region within the image, and the agent is required to accurately localize the bounding box corresponding to the referred expression. In VQA, the agent must reason over visual input and answer natural language questions. We evaluate both in the standard setting (*e.g.*, VQAv2) and in instruction-following setups. In VE, the agent needs to determine whether a textual hypothesis (*e.g.*, a statement, a question, or a sentence) is entailed, contradicted, or neutral with respect to an image. In IC, the agent generates fluent natural language captions that describe the visual content. For all tasks, we pretrain models using EC tokens in place of natural language and then fine-tune them on distinct sets of downstream instances with natural language annotations to assess generalization and cross-modal reasoning performance.

## 4  Experimental Setup

We experiment with a unified vision-language model (VLM) capable of handling a range of VL tasks. Specifically, we adopt OFA (Wang et al., 2022a), a Transformer-based architecture designed for both generation and classification tasks. OFA encodes inputs as sequences of discrete tokens from a unified vocabulary, enabling seamless processing of visual, linguistic, and other modalities. The model is pre-trained on multimodal VL datasets, including visual grounding and visual question answering, as well as several unimodal datasets,

including image infilling, object detection, and text infilling. These tasks allow the model to develop informative cross-modal representations for relevant input modalities. Despite its generative nature, OFA also supports classification tasks like VQA by constraining generation to a candidate answer set (Wang et al., 2022a). For grounding, it directly predicts bounding box coordinates, avoiding reliance on region proposals. This unified modeling approach makes OFA an ideal backbone for evaluating the transfer effects of emergent communication (EC) pretraining. We evaluate OFA on four standard VL benchmarks, RefCOCO+ (VRE), VQAv2 (VQA), SNLI-VE (VE), and MSCOCO (IC), under three settings: (1) Zero-shot baseline using the released OFA base model without additional pretraining or fine-tuning; (2) EC pretraining, where we pretrain OFA on EC-labeled versions of each dataset before fine-tuning on natural language annotations; and (3) Natural language (NL) pretraining, where we pretrain on the original captioned datasets before fine-tuning. We utilize the officially released OFA base model weights and checkpoints. The zero-shot results serve as a lower bound, allowing us to isolate the contribution of EC pretraining in improving downstream task performance. To further examine the scalability of EC pretraining, we extend our experiments to LLaVA-1.5 (Liu et al., 2024a), a state-of-the-art VLM. LLaVA-1.5 is trained in two stages: vision-language feature alignment, connecting a CLIP-based vision encoder (Radford et al., 2021) to a frozen Vicuna LLM (Chiang et al., 2023), followed by visual instruction tuning. We construct LLaVA-1.5-EC, a variant in which natural language captions in the pretraining corpus are replaced with EC tokens generated by our trained speaker. The standard LLaVA-1.5 fine-tuning pipeline is retained, allowing us to assess how EC-based representations influence performance on instruction-following benchmarks. Additional implementation details can be found in Appendix A.

## 4.1 Datasets

**Visual Referring Expression (VRE).** We evaluate on the standard RefCOCO benchmark suite (Yu et al., 2016), which includes RefCOCO, RefCOCO+, and RefCOCOg, all derived from the MS-COCO image dataset (Lin et al., 2014). These benchmarks contain natural language expressions referring to specific objects or regions within images, collected through human annotation (Kazemzadeh et al., 2014; Yu et al., 2016). RefCOCO contains 142,209 expressions for 50,000 objects across 19,994 images. RefCOCO+ includes 141,564 expressions for 49,856 objects and focuses on appearance-based descriptions by filtering out location-based cues. Each dataset is split into three subsets: val, testA, and testB, where testA contains images with people and testB consists of non-person images, enabling more fine-grained evaluation. We use the OFA-preprocessed version of these datasets (Wang et al., 2022a), where each referring expression is paired with a unique bounding box to standardize model input and supervision.

**Visual Question Answering (VQA).** We conduct experiments on the VQAv2 dataset (Goyal et al., 2017). VQAv2 comprises a collection of approximately 1.1M samples from $200,000$ images accompanied by 13M natural language answers (Goyal et al., 2017). As per OFA, instances for which multiple potential answers exist for a given question-image pair were split into individual samples for each potential answer (Wang et al., 2022a). This OFA-adapted version of the VQAv2 dataset includes training, validation, and test sets, with ∼1.8M, 10,402, and 447,793 samples, respectively. To enable EC pretraining, we divide the training set into two halves. For the first half, we replace the original natural language answers with emergent communication (EC) tokens generated by the speaker model, producing an EC-labeled corpus. The second half retains the original human annotations and is used for fine-tuning the model on the standard VQA task. This setup allows us to assess how well EC pretraining transfers to natural language question answering.

**Visual Entailment (VE).** In VE, an image $\mathbf{I}$ serves as a premise along with a hypothesis text $\mathbf{H}_{\text{text}}$. The objective is to determine whether the hypothesis can be inferred from the image, *i.e.*, if the image entails the hypothesis (Xie et al., 2019). The model must classify the pair into one of three categories: *entailment* (if there is sufficient evidence in $\mathbf{I}$ to conclude that $\mathbf{H}_{\text{text}}$ is true), a *contradiction* (if there is enough evidence in $\mathbf{I}$ to conclude that $\mathbf{H}_{\text{text}}$ is false), or *neutral* (if there is insufficient evidence in $\mathbf{I}$ to make a conclusion about $\mathbf{H}_{\text{text}}$). VE shares similarities with VQA as both tasks require the model to reason and make logical deductions based on an image. However, VE is more complicated than VQA, in the sense that choosing whether a text entails an image requires complex fine-grained reasoning beyond answering a question. We evaluate VE performance using the SNLI-VE dataset (Xie et al., 2019; 2018), which contains 529,527 training samples, 17,858 validation samples, and 17,901 test samples derived from 29,783 unique images. To examine the effect

of EC pretraining under different supervision levels, we conduct experiments using the full training set as well as reduced subsets of 50,000 and 10,000 samples. These settings allow us to assess the generalization benefits of EC across varying data regimes.

**Image Captioning (IC).** We evaluate image captioning performance using the Microsoft COCO dataset (Lin et al., 2014), a widely used benchmark for vision-language tasks. Image captioning involves generating fluent, semantically rich captions that accurately describe input images. We adopt the OFA-preprocessed version of MSCOCO, which is split into four subsets: caption_stage1_train, caption_stage2_train, caption_val, and caption_test. Each image in caption_stage1_train is paired with a single caption, while images in the other subsets are associated with approximately five captions each. The dataset includes ∼566K samples in caption_stage1_train, 113K in caption_stage2_train, 5K in the validation set, and 5K in the test set. For evaluation, we follow the Karpathy split (Karpathy & Fei-Fei, 2015) and report results using established automatic metrics: BLEU-4 (Papineni et al., 2002), METEOR (Banerjee & Lavie, 2005), CIDEr (Vedantam et al., 2015), and SPICE (Anderson et al., 2016).

**Instruction-Following Benchmarks.** We adopt the LlaVA-1.5 pretraining and fine-tuning dataset, with modified captions from natural language to EC tokens during pretraining. In LlaVA-1.5, pretraining utilizes approximately 558K images sampled from the LAION (Schuhmann et al., 2022), CC (Changpinyo et al., 2021), and SBU (Ordonez et al., 2011) synthesized captioning datasets, where captions are reformatted into simple instruction-following prompts. To construct the instruction-following dataset, natural image-text pairs are augmented with human-issued instructions. In the visual instruction tuning stage, we fine-tune the model end-to-end on a mixture of ∼665K multimodal instruction-following examples, synthesized and sampled from a variety of VQA data sources, including GPT-generated content, GQA (Hudson & Manning, 2019), COCO (Lin et al., 2014), and TextVQA (Singh et al., 2019). In addition, we evaluate on multimodal benchmarks POPE (measuring hallucination resilience) Li et al. (2023b), MME (visual perception) Fu et al. (2023), and MM-Vet (visual conversations) Yu et al. (2024).

## 4.2   EC Pre-training

**Pretraining on Visual Referring Expression (VRE).** We pre-train the OFA base model for VRE on the RefCOCO train set. We use the EC-generated text from the referential game speaker agent as captions for pre-training. To obtain the EC text, we employ a ResNet-18 model (He et al., 2016) and extract 512-dimensional image features from each image in the RefCOCO training set. These image features are then passed to a speaker agent that generates a set of Emergent Communication (EC) tokens describing the image. Finally, we replace the original natural language RefCOCO caption that describes a bounding box with the EC-generated tokens. As the EC tokens encompass information about the image, we hypothesize that they implicitly capture information relevant to the associated bounding box, thus offering potential pre-training benefits. The resulting RefCOCO dataset, post-processed with EC tokens as captions, serves as the basis for pre-training the OFA base VRE model.

**Pretraining on Multiple Tasks.** To assess the broader utility of EC tokens, we conduct multitask pretraining on a unified dataset combining Visual Referring Expression (RefCOCO), Image Captioning (MSCOCO), and Visual Question Answering (VQAv2). We construct two versions of this dataset: one using natural language (NL) annotations and another using EC-generated tokens in place of all NL captions and answers. EC text is generated using the same pipeline as in the VRE experiment, where ResNet-18 features are passed to the speaker agent to produce discrete messages that replace the original text with these EC-generated tokens. After pretraining, we fine-tune the best-performing checkpoints for both the EC and NL-pretrained models independently on downstream vision-language tasks: Visual Referring Expression (VRE) and Visual Entailment (VE). The baseline is an OFA model with randomly initialized weights trained from scratch on each task without any pretraining. These experiments aim to elucidate the added value of EC text in comparison to models trained without pretraining or those pretrained solely with NL data, thus offering a comprehensive understanding of the role of emergent language in enhancing vision-language tasks.

**Instruction-Following Tasks.** We benchmark the effectiveness of EC pretraining on instruction-following tasks using LLaVA-1.5 (Liu et al., 2024a), a state-of-the-art, open-source VLM. LLaVA-1.5 integrates a CLIP-based vision encoder (Radford et al., 2021) with a Vicuna language model (Chiang et al., 2023) and

is trained end-to-end to handle both visual and language modalities, achieving GPT-4-level performance on multimodal chat benchmarks. LLaVA-1.5 is trained in two stages: (1) a feature alignment stage, where the CLIP visual encoder is aligned with the Vicuna LLM via image-caption pairs, and (2) a visual instruction tuning stage using a diverse set of vision-language instruction-following data. In the original LLaVA-1.5, the alignment phase uses ∼558K image-text pairs drawn from LAION (Schuhmann et al., 2022), Conceptual Captions (Changpinyo et al., 2021), and SBU (Ordonez et al., 2011), with captions refined into simplified instruction templates to balance conceptual coverage. To evaluate the utility of EC tokens in this high-capacity setting, we introduce **LLaVA-1.5-EC**, a variant trained using the same images from LAION-CC-SBU but replacing all natural language captions with EC sequences. These EC tokens are generated using our trained EC speaker models, which produce discrete symbolic messages per image. The resulting EC-based dataset is then used for the entire alignment phase in place of natural language captions. We hypothesize that these emergent tokens, though artificial, capture visually grounded and compositional semantics sufficient to train a competitive instruction-following VLM.

### 4.3 NL Fine-tuning

**Visual Referring Expression (VRE).** We fine-tune the best-performing EC-pretrained VRE model checkpoint on RefCOCO+. By using different visual grounding datasets in pre-training and fine-tuning stages (RefCOCO for pre-training and RefCOCO+ for fine-tuning), we ensure that information does not overlap between the two phases. Subsequently, we evaluate the performance of the fine-tuned model on all Ref-COCO+ splits (val, testA, and testB). For NL fine-tuning, we fine-tune the OFA base model on the VRE task, employing natural language captions from the RefCOCO+ dataset. We consider this as an upper-bound model for comparison against the EC pretraining experiment. Following fine-tuning, we evaluate the fine-tuned model's performance on all RefCOCO+ validation and test sets.

**Visual Question Answering (VQA).** To examine the performance of EC pretraining on the Visual Question Answering task, we considered the Unified EC model obtained after pretraining on multiple tasks detailed in Section 4.2. We fine-tune the best-performing Unified models (both EC and NL) on half of the VQAv2 dataset with natural language question-answer pairs. The other half was used during pretraining. This helps avoid data overlap and overfitting. We consider NL fine-tuning as an upper bound for comparison against EC pretraining. Subsequently, we evaluate the fine-tuned models on the VQAv2 validation and test-dev sets.

**Visual Entailment (VE).** To assess the potential of emergent communication (EC) models for transferring knowledge to new vision language tasks, we employ the EC pre-trained model obtained from VRE as pre-training for the VE task. We fine-tune the best EC pre-trained model checkpoint on the SNLI-VE training set and evaluate on the SNLI-VE test and dev validation sets. To further understand the impact of EC under varying sampling conditions, we conduct fine-tuning on randomly sampled sets with different sizes, *e.g.*, 10,000 and 50,000 samples. For the NL experiment, we fine-tune the OFA base model on SNLI-VE. To assess the impact of training set sample size on model performance, we similarly perform fine-tuning with different training set sample sizes and evaluate our models on the SNLI-VE validation and test sets.

**Image Captioning (IC).** To further assess the transferability of EC pretraining, we evaluate performance on the MS COCO image captioning benchmark (Lin et al., 2014). Specifically, we fine-tune the best-performing EC-pretrained model – initially trained on RefCOCO for visual referring expressions – on the MSCOCO captioning task. This experiment probes whether EC-based grounding can generalize to open-ended image description generation. For comparison, we fine-tune the OFA base model directly on MSCOCO using natural language (NL) captions, treating it as an upper-bound reference. Evaluation is performed on the MS COCO test split using standard automatic metrics: BLEU-4 (Papineni et al., 2002), METEOR (Banerjee & Lavie, 2005), CIDEr (Vedantam et al., 2015), and SPICE (Anderson et al., 2016).

**Instruction-Following Tasks.** The visual instruction tuning phase involves end-to-end fine-tuning on a curated mixture of approximately 665k multimodal instruction-following samples from VQA datasets such as GQA (Hudson & Manning, 2019), COCO (Lin et al., 2014), and TextVQA (Singh et al., 2019), as well as GPT-generated instruction-following content. This phase is designed to align the model's responses with natural language instructions grounded in visual input. In our LLaVA-1.5-EC model, we adopt the same instruction tuning protocol used in the original LLaVA-1.5 framework (Liu et al., 2024a). The only

modification lies in the pretraining phase, where we substitute natural language captions with discrete EC tokens. To benchmark the effectiveness of EC-based pretraining, we compare against the original LLaVA-1.5 model trained entirely with natural language and various mainstream open-source VLMs, including BLIP-2 (Li et al., 2023a), Instruct-BLIP (Dai et al., 2023), Shikra (Chen et al., 2023), IDEFICS (IDEFICS, 2023), and Qwen-VL (Bai et al., 2023).

## 5 Experimental Results

### 5.1 Visual Referring Expression (VRE)

Table 1 reports accuracy on the RefCOCO+ Visual Referring Expression (VRE) task across three configurations: **Base** (OFA without pretraining), **+EC** (EC-pretrained, then fine-tuned on NL), and **+NL** (NL-pretrained and fine-tuned on NL). Pretraining on EC tokens yields a substantial improvement over the Base model, increasing accuracy by over $2\times$ on all splits. This demonstrates the strong transferability of EC tokens, which are generated without any natural language supervision and yet encode rich semantic structure grounded in visual input. While

Table 1: **VRE Accuracy** on RefCOCO+ splits. **Base**: OFA without pretraining. **+EC**: EC-pretrained, then NL fine-tuned. **+NL**: NL-pretrained and fine-tuned.

| Model | val | testA | testB |
|---|---|---|---|
| **Base** | 29.81 | 31.49 | 27.53 |
| **+EC** | 62.17 | 67.15 | 51.16 |
| **+NL** | 81.91 | 86.60 | 73.49 |

NL pretraining still achieves the highest scores, which is expected due to the compositional and grounded nature of natural language, EC pretraining reaches approximately two-thirds of the performance of NL pretraining across all evaluation splits. This finding highlights the potential of EC-based pretraining in low-resource or label-scarce settings, where collecting large-scale image-text pairs is infeasible. Notably, the EC referential game and the generation of EC tokens by the speaker agent can be performed on unlabeled images in the wild, effectively producing descriptive captions for these images. Consequently, EC tokens contain valuable semantic information about the images, which can prove advantageous for vision-language pretraining tasks. These findings emphasize the potential of leveraging EC pre-training for vision-language tasks. A deeper qualitative analysis in Appendix B reveals that EC tokens exhibit latent compositional syntax, polysemy, and prosodic-like structures, often paralleling natural language in their ability to encode semantics, disambiguate meaning through repetition, and refine concepts via positional cues.

### 5.2 Pretraining on Multiple Tasks

Table 2 demonstrates the effectiveness of EC pretraining as a general-purpose initialization for multimodal tasks. In the context of VRE, EC pretraining surpasses a baseline model trained directly on visual grounding tasks without additional pretraining, achieving an impressive relative gain of over 108%. Similarly, in VQA, EC improves test-dev accuracy by 11.5%. These gains highlight the ability of EC tokens, learned without human annotation, to transfer semantic and structural information across vision-language tasks. While NL pretraining still leads to the highest overall performance, EC pretraining achieves notable improvements despite

Table 2: **Unified Pretraining.** EC pretraining yields consistent gains. **Base**: OFA without pretraining. **+EC**: EC-pretrained, then NL-finetuned. **+NL**: NL-pretrained and NL-finetuned.

| Model | VRE (RefCOCO+) | | | VQA (VQAv2) | |
|---|---|---|---|---|---|
| | val | testA | testB | val | test-dev |
| **Base** | 10.03 | 13.88 | 9.72 | 49.33 | 40.80 |
| **+EC** | 23.77 | 28.89 | 18.84 | 50.61 | 45.49 |
| **+NL** | 40.15 | 45.90 | 31.34 | 56.66 | 51.26 |

not relying on natural language supervision. These findings suggest that EC-based pretraining can be a viable alternative or complementary strategy in settings with limited access to natural language annotations, enabling more scalable and generalizable vision-language learning.

### 5.3 Visual Entailment (VE)

Figure 2 presents results for the visual entailment task using models initialized from EC-VRE pretraining. Across varying training set sizes, EC pretraining consistently improves VE performance over the baseline model. These results provide compelling evidence of the utility of EC tokens for transferring knowledge

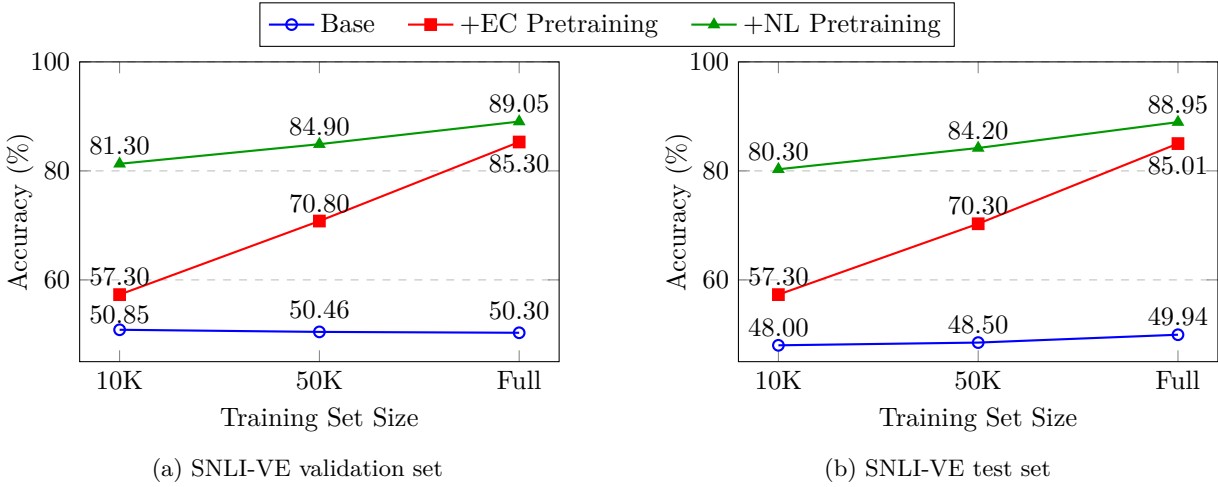

(a) SNLI-VE validation set

(b) SNLI-VE test set

Figure 2: **Visual Entailment (VE) Accuracy with Varying Training Sizes.** EC pretraining substantially improves VE accuracy compared to the baseline across all training sizes, and approaches NL pretraining performance as more downstream data becomes available. **Base**, **+EC Pretraining**, and **+NL Pretraining**, corresponding to training no pretraining, EC pretraining followed by natural language (NL) fine-tuning, and NL pretraining followed by NL fine-tuning.

Table 3: **Image Captioning Accuracy.** +EC model consistently outperforms the base model on all 8 metrics, often doubling or quadrupling performance. **Base**: OFA without pretraining. **+EC**: EC-pretrained then NL fine-tuned. **+NL**: NL-pretrained then NL fine-tuned.

| Model | BLEU-1 | BLEU-2 | BLEU-3 | BLEU-4 | METEOR | ROUGE | CIDEr | SPICE |
|-------|--------|--------|--------|--------|--------|-------|-------|-------|
| **Base** | 48.70 | 26.58 | 14.92 | 8.45 | 12.11 | 35.65 | 0.12 | 3.94 |
| **+EC** | 70.54 | 52.46 | 38.08 | 27.39 | 22.77 | 50.95 | 0.83 | 15.78 |
| **+NL** | **75.15** | **58.60** | **44.13** | **32.71** | **26.31** | **54.98** | **1.06** | **19.76** |

between VL tasks. When comparing EC pre-training with the natural language fine-tuning of OFA, it is expected that the latter would achieve better performance. However, it is surprising to observe that EC pre-training can yield comparable performance in certain settings. For instance, when fine-tuned on the full training data, EC achieves an accuracy of 85.01% on the test set, whereas NL achieves 88.95%. The relatively small performance gap between EC-VRE pre-training and natural language fine-tuning highlights the transferability benefits of EC pre-training and opens up future directions in utilizing emergent language for complex vision language tasks and more efficient VL modeling.

## 5.4 Image Captioning (IC)

To further evaluate the transferability of EC pretraining, we fine-tune the EC-VRE model – originally pretrained on RefCOCO – on an image captioning task. As shown in Table 3, EC pretraining yields substantial improvements across all standard evaluation metrics. Most notably, it achieves more than a 4× increase in CIDEr score compared to the baseline model and more than doubles performance on several other metrics. Despite having no access to human-written captions during pretraining, the EC-pretrained model approaches the performance of models initialized with natural language (NL) pretraining. These results underscore the representational strength of EC tokens for open-ended generation tasks and point to their potential utility in multimodal learning scenarios with limited supervision.

## 5.5 Instruction-Following Tasks

Table 4 compares LLaVA-1.5-EC, which is trained exclusively on EC tokens, against a range of state-of-the-art VLMs across five benchmarks: VQAv2 (Goyal et al., 2017), GQA (Hudson & Manning, 2019), VizWiz

Table 4: **Comparing LLaVA-1.5-EC with SoTA Instruction-Following Models.** LLaVA-1.5-EC, pretrained solely on EC tokens, achieves strong performance across five diverse VQA tasks, despite not using natural language during pretraining. While slightly trailing LLaVA-1.5 on VQAv2 and GQA, it matches or outperforms BLIP-2 and InstructBLIP in most settings. Results highlight the transferability and robustness of EC-based pretraining for downstream multimodal reasoning tasks. *Training images/annotations of the datasets are observed during training. †Includes in-house data that is not publicly accessible.

| Method | LLM | Image Size | Sample Size Pretrain | Finetune | VQAv2 | GQA | VizWiz | SciQA IMG | TextVQA |
|---|---|---|---|---|---|---|---|---|---|
| BLIP-2 | Vicuna-13B | $224^2$ | 129M | - | 65.0 | 41 | 19.6 | 61 | 42.5 |
| InstructBLIP | Vicuna-7B | $224^2$ | 129M | 1.2M | – | 49.2 | 34.5 | 60.5 | 50.1 |
| InstructBLIP | Vicuna-13B | $224^2$ | 129M | 1.2M | – | 49.5 | 33.4 | 63.1 | 50.7 |
| Shikra | Vicuna-13B | $224^2$ | 600K | 5.5M | 77.4* | – | – | – | – |
| IDEFICS-9B | LLaMA-7B | $224^2$ | 353M | 1M | 50.9 | 38.4 | 35.5 | – | 25.9 |
| IDEFICS-80B | LLaMA-65B | $224^2$ | 353M | 1M | 60.0 | 45.2 | 36.0 | – | 30.9 |
| Qwen-VL | Qwen-7B | $448^2$ | 1.4B† | 50M† | 78.8* | 59.3* | 35.2 | 67.1 | 63.8* |
| Qwen-VL-Chat | Qwen-7B | $448^2$ | 1.4B* | 50M† | 78.2* | 57.5* | 38.9 | 68.2 | 61.5* |
| **LLaVA-1.5** | Vicuna-7B | $336^2$ | **558K** | **665K** | 78.5* | 62.0* | 50.0 | 66.8 | 58.2 |
| **LLaVA-1.5-EC** | Vicuna-7B | $336^2$ | **558K** | **665K** | 72.03* | 55.27* | 40.03 | 64.45 | 48.61 |

Table 5: **Evaluation on Multimodal Benchmarks.** LLaVA-1.5-EC-7B demonstrates strong robustness, outperforming InstructBLIP-14B on the POPE (adv) subset, MME, and MM-Vet, despite lacking natural language-aligned pretraining, highlighting EC's potential as a scalable and compositional grounding signal.

| Method | POPE (rand) | POPE (pop) | POPE (adv) | MME | MM-Vet |
|---|---|---|---|---|---|
| BLIP2-14B | 89.6 | 85.5 | 80.9 | 1293.8 | 22.4 |
| InstructBLIP-14B | 87.7 | 77.0 | 72.0 | 1212.8 | 25.6 |
| LLaVA-7B | 76.3 | 72.2 | 70.1 | 809.6 | 25.5 |
| LLaVA-1.5-CLIP-7B | 17.1 | 17.1 | 17.1 | 687 | 11.3 |
| **LLaVA-1.5-7B** | 87.3 | 86.1 | 84.2 | 1510.7 | 31.1 |
| **LLaVA-1.5-EC-7B** | 75.1 | 74.6 | 74.0 | 1274.4 | 27.8 |

(Gurari et al., 2018), SciQA-IMG (Lu et al., 2022), and TextVQA (Singh et al., 2019). Despite using only 558K images and no natural language supervision, LLaVA-1.5-EC achieves competitive performance, outperforming well-established models such as BLIP-2 (13B) and InstructBLIP (13B) across most datasets. For instance, relative to BLIP-2, LLaVA-1.5-EC achieves a 104.23% gain on VizWiz, 34.8% on GQA, and 10.8% on VQAv2. It also outperforms InstructBLIP, which is trained on 129M captioned image-text pairs and fine-tuned on 1.2M additional examples, highlighting the impressive representational capacity of EC pretraining. While LLaVA-1.5-EC does not surpass Qwen-VL – trained with over a billion curated image-text pairs – it achieves strong results despite having seen 2–3 orders of magnitude fewer samples and no human-written captions. For example, on VizWiz, LLaVA-1.5-EC even outperforms Qwen-VL (40.03 vs. 35.2), showing its effectiveness on visually grounded tasks.

Table 5 demonstrates that LLaVA-1.5-EC-7B outperforms InstructBLIP-14B on multimodal benchmarks POPE (adv), MME, and MM-Vet, despite lacking any natural language supervision during pretraining. These results highlight the semantic precision and transferability of EC tokens under distribution shifts and reinforce the hypothesis that EC pretraining induces grounded, compositional structures that are well-aligned with real-world vision-language reasoning demands. While LLaVA-1.5 still outperforms EC in absolute terms (as expected due to the use of curated human language), the relatively small performance gap highlights the surprising generalization capacity of emergent communication and its potential as a scalable pretraining strategy in low-resource or text-scarce settings. Results so far also reveal that emergent language may encode compositional and semantic patterns that are beneficial for downstream reasoning, a hypothesis we investigate more deeply in Section 5.6 through targeted ablation studies.

In addition, Figure 3 compares LLaVA-1.5-EC against leading VLMs on MMBench (Liu et al., 2024b), a rigorous benchmark suite for evaluating instruction-following and multimodal reasoning in VLMs. Both MMBench (EN) and its Chinese-translated variant (MMBench-CN) include a broad set of multimodal ques-

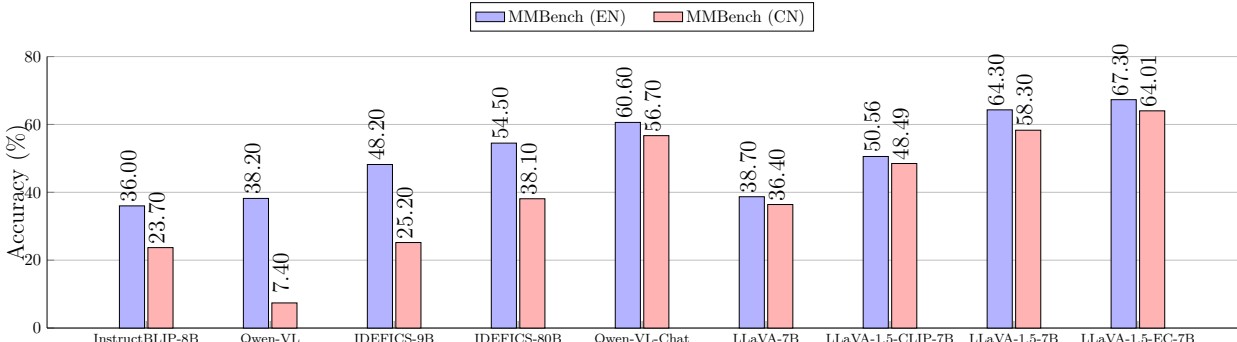

Figure 3: **Comparison of LLaVA-1.5-EC with SoTA Instruction-Following Models on the MM-Bench Benchmark (English and Chinese).** LLaVA-1.5-EC, pretrained using Emergent Communication tokens, surpasses all baselines, highlighting the potential of EC-based pretraining.

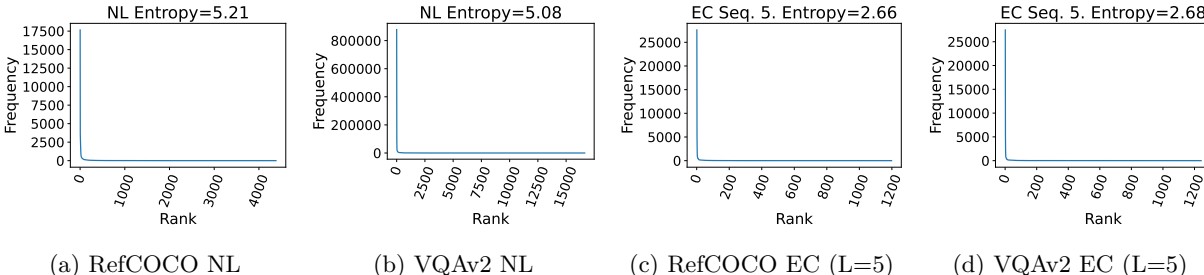

(a) RefCOCO NL  (b) VQAv2 NL  (c) RefCOCO EC (L=5)  (d) VQAv2 EC (L=5)

Figure 4: **Unigram distributions for Natural Language (NL) and EC tokens (L=5).** Each plot shows token frequency ranks for the first 10K samples from RefCOCO and VQAv2 datasets. Compared to NL, EC token distributions are more peaked and concentrated on fewer symbols, resulting in lower entropy (RefCOCO: 2.66 vs 5.21; VQAv2: 2.68 vs 5.08). This suggests that shorter EC messages rely on fewer reused tokens to convey meaning efficiently, while the versatility of natural language allows it to incorporate way more unique tokens in comparison.

tions across diverse topics, assessing a model's capacity to follow instructions, ground language in visual input, and make fine-grained distinctions. Despite being trained exclusively on discrete EC tokens without any natural language captions, our 7B parameter LLaVA-1.5-EC model achieves the highest accuracy across both English (67.3%) and Chinese (64.01%) MMBench sets. It outperforms significantly larger models such as InstructBLIP-8B, IDEFICS-80B, and Qwen-VL, the latter trained on over a billion in-house images. Notably, LLaVA-1.5-EC consistently outperforms a LLaVA-CLIP baseline, which directly fine-tunes LLaVA-1.5 with raw CLIP features, demonstrating that EC-based pretraining provides a more semantically structured and transferable representation than direct projection from CLIP features alone. Even more impressively, LLaVA-1.5-EC surpasses both the original LLaVA-7B and the improved LLaVA-1.5-7B models, despite relying on entirely synthetic pretraining. These results underscore the strength of EC-based pretraining when combined with LLaVA-1.5's visual instruction tuning, and demonstrate the potential of emergent language as an effective and scalable pretraining signal for multimodal reasoning tasks.

## 5.6  Ablations

We conduct ablations to analyze the statistical and structural properties of EC tokens and identify factors driving their transferability. Specifically, we aim to address two central questions: (1) To what extent are EC tokens consistent and meaningful across tasks? and (2) Does the structure of EC messages, beyond token identity, contribute to transfer performance? To this end, we analyze unigram distributions to study vocabulary usage across datasets and conduct word-order perturbation experiments to probe the role of compositional structure in EC representations.

Table 6: **Effect of Structure and Semantics in EC Language.** $EC_{orig}$, $EC_{reordered}$, $EC_{random}$ represent the original EC language generated by the speaker, perturbed EC language by reshuffling, and random EC tokens, respectively. All models are fine-tuned on RefCOCO+. $EC_{orig}$, the model pre-trained on the original EC language, exhibits higher downstream performance on all evaluation splits compared to the models pre-trained on shuffled EC language and random EC language. Results confirm that the structure and token semantics of EC language play a critical role in its transferability for vision-language tasks.

| | Zero-Shot | $EC_{orig}$ | $EC_{reordered}$ | $EC_{random}$ |
|---|---|---|---|---|
| **val** | 29.81 | **42.50** | 42.10 | 42.10 |
| **testB** | 27.53 | **34.49** | 33.63 | 33.34 |

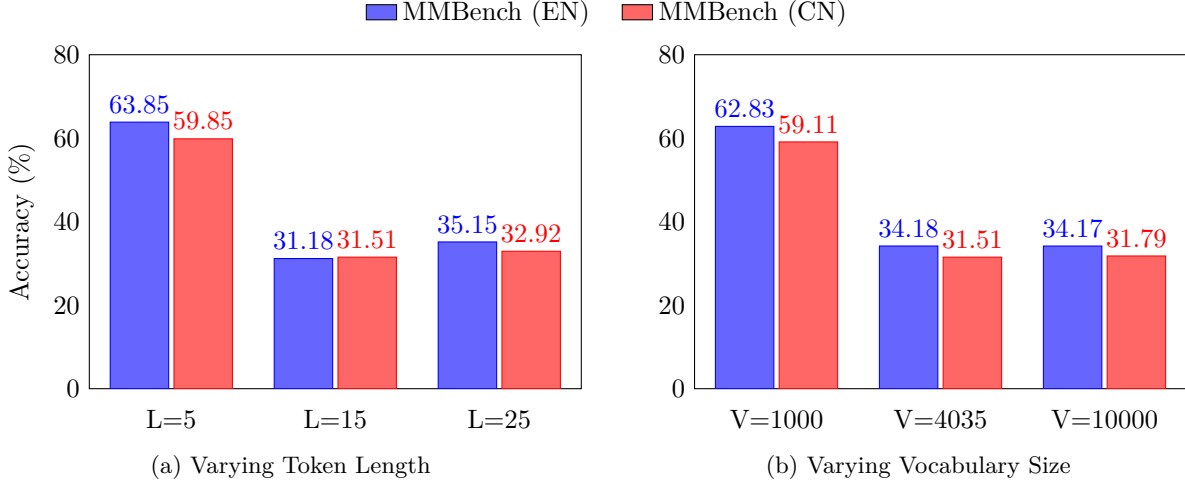

(a) Varying Token Length

(b) Varying Vocabulary Size

Figure 5: **Ablation Studies on EC Token Design.** (Left) Accuracy decreases with longer token sequences. (Right) Smaller vocabularies improve generalization.

**Unigram Distributions.** We compare the unigram distributions of natural language (NL) and emergent communication (EC) across RefCOCO and VQAv2. For EC, we generate sequences of fixed lengths (L={5, 15, 25}) from the first 10K training samples using a trained speaker model. All text is tokenized using the NLTK word tokenizer[1], and unigram counts are computed and sorted to generate the respective unigram distributions for each corpus. Figure 4 unigram distribution plots reveal notable differences in lexical diversity between natural language (NL) and emergent communication (EC) corpora. For both RefCOCO and VQAv2 datasets, NL sequences show substantially higher entropy (5.33 and 5.08 for RefCOCO and VQAv2, respectively), indicating broader token usage and greater linguistic variability. In contrast, EC texts generated with fixed length L=5 exhibit lower entropy (4.52 for RefCOCO, 4.63 for VQAv2), suggesting a more concentrated and repetitive token usage. This implies that EC, despite being artificially generated, converges on a smaller, more structured vocabulary that is reused efficiently, likely to optimize communication within tight length constraints. The more peaked EC distributions further support the idea that emergent messages encode semantics compactly, relying on structure and positional encoding rather than lexical diversity.

**Investigating the Importance of Structure in EC Tokens.** To better understand the role of structure and semantics in EC language, we conduct an ablation study using three variations of EC pretraining data: (1) **$EC_{orig}$**: the original EC tokens generated by the referential game speaker, (2) **$EC_{reordered}$**: EC tokens with their order randomly shuffled, and (3) **$EC_{random}$**: sequences of randomly sampled tokens of matching length. Each variant is used to pretrain a task-agnostic OFA model on a 30K subset of RefCOCO. After pretraining, we fine-tune each model on 10K samples of RefCOCO+ and evaluate on standard validation and test splits. Results in Table 6 show that **$EC_{orig}$** yields the best downstream performance, highlighting the

---

[1]https://www.nltk.org/api/nltk.tokenize.html

Table 7: **EC Token Entropy and Referential Game Success Rate** ablations varying token length (left, vocab size fixed at 4035) and vocabulary size (right, token length fixed at 15).

| Token Len | Entropy | Success Rate (%) | Vocab Size | Entropy | Success Rate (%) |
|---|---|---|---|---|---|
| 5 | 2.68 | 77.31 | 1,000 | 2.45 | 81.46 |
| 15 | 3.82 | 95.59 | 4,035 | 3.82 | 98.39 |
| 25 | 4.58 | 99.27 | 10,000 | 2.21 | 86.88 |

importance of both token order and semantic content in emergent communication. These findings suggest that EC tokens encode nontrivial structure and meaning that support generalization in vision-language tasks.

**Channel Capacity–Generalization Tradeoff in EC Token Design.** We further examine how the communication capacity of emergent language, defined by token sequence length and vocabulary size, impacts both intra-agent coordination and downstream transfer. Token length and vocabulary size jointly determine the theoretical channel capacity (measured as $\log_2(V) \times L$), *i.e.*, the maximum information an agent can convey per message. For each configuration, $L \in \{5, 15, 25\}$ and $V \in \{1000, 4035, 10000\}$, we pretrain a LLaVA-EC model using the resulting EC corpus and evaluate both (1) downstream accuracy on MMBench and (2) channel utilization via token entropy and referential game success rate.

As shown in Table 7, increasing token length consistently increases both message entropy and referential accuracy, indicating greater utilization of the available capacity. However, this higher utilization correlates with lower downstream accuracy (Figure 5a), suggesting that agents are encoding redundant messages optimized for referential discrimination but poorly aligned with transferable semantic abstractions. The shortest configuration ($L = 5$) performs best downstream, highlighting the benefit of a compressed emergent protocol that emphasizes abstraction over specificity. A similar trend is observed when varying vocabulary size (Figure 5b, Table 7): the smallest vocabulary ($V = 1000$) yields the highest downstream accuracy, despite lower entropy and referential success. In contrast, larger vocabularies result in higher entropy and referential performance but worse transfer, consistent with over-specialization or degenerate symbol use.

These results collectively reveal a channel capacity–generalization tradeoff in emergent communication. While greater capacity improves intra-agent coordination, as evidenced by higher entropy and referential success, it also increases the risk of overfitting, symbolic drift, and protocol inefficiency. This aligns with findings by Chaabouni et al. (2019), who observed that agents tend to produce overly long and inefficient messages in unconstrained settings. Our results extend these insights by demonstrating that capacity bottlenecks, introduced via shorter sequences or smaller vocabularies, encourage symbolic reuse and abstraction, ultimately yielding better generalization to downstream multimodal tasks.

## 6 Conclusion

This work investigates the potential of Emergent Communication (EC) as a learning signal for vision-language models. We explore whether discrete messages generated by referential game agents can serve as effective supervision for cross-modal pretraining. Through extensive experiments across diverse tasks, including visual referring expression, visual question answering, visual entailment, and image captioning, we demonstrate that EC pretraining significantly improves performance over strong non-pretrained baselines and, in some cases, approaches or surpasses models trained with natural language. We further validate EC's generalization by scaling to instruction-following tasks, demonstrating that LLaVA-1.5-EC achieves competitive accuracy on instruction-following benchmarks despite being trained solely on synthetic EC tokens. Ablations reveal that EC tokens encode structural and semantic signals crucial for transfer learning. These findings suggest that EC can serve as a scalable and complementary resource for vision-language pretraining, particularly in low-resource or weakly supervised settings.

One promising direction is to develop a new class of pretraining objectives grounded entirely in interaction, *i.e.*, replacing static captioning supervision with agent-driven communication. Another exciting possibility is to evolve EC into a fully expressive representational language that can interface directly with down-

stream models, prompting them as effectively as human language. Further, future research could explore the emergence of syntax or compositional grammar within EC tokens and whether EC can self-organize into hierarchies aligned with semantic or spatial structures in images. Finally, we envision agent ecosystems where EC evolves dynamically over time through open-ended interaction with multimodal environments, offering a path toward continual, self-supervised vision-language learning without reliance on human annotations.

## 7 Acknowledgments

This research is based on work partially supported by the Amazon–Virginia Tech Initiative for Efficient and Robust Machine Learning and by the U.S. Defense Advanced Research Projects Agency (DARPA) under award numbers HR00112390062 and HR001125C0303. The views and conclusions contained herein are those of the authors and should not be interpreted as necessarily representing the official policies, either expressed or implied, of Amazon, DARPA, or the U.S. Government. The U.S. Government is authorized to reproduce and distribute reprints for governmental purposes notwithstanding any copyright annotation therein.

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

# A    Implementation Details

The EC speakers used to generate the EC datasets are directly trained on the COCO image features from the codebase[2] of Yao et al. (2022) for 2000 epochs. Training is performed on a P100 GPU, and the sequence length limit is set to 15. Generating EC sequences of length 15, the speakers draw from a vocabulary size of 4035 tokens. In the Visual Referring Expression (VRE) experiments, we adopt the codebase of OFA[3] (Wang et al., 2022a) and mostly follow their default setup. Initially, we pre-train the OFA model on the RefCOCO training set with EC annotations via continuous pretraining (Wang et al., 2022a). The pretraining data was prepared to align with OFA's structure, as outlined on their GitHub page. The pretraining process consists of 17 epochs and 492,000 updates. Subsequently, we fine-tune the pre-trained model for 10 epochs and 18,500 updates. Pretraining was conducted on a single NVIDIA A100 GPU for 2 days, while the fine-tuning phase required 2 NVIDIA A100 GPUs and took 2 days to complete. For the Visual Question Answering (VQA) task, we again adopt the OFA codebase and follow the default setup provided in both the continuous pretraining and VQA fin-tuning and evaluation scripts. Continuous pretraining was executed on a single NVIDIA V100 GPU for 4 days, encompassing 960,000 updates, which corresponded to approximately 4 to 5 epochs. Fine-tuning for 5 epochs required around 90 hours on 2 P100 GPUs. In the Visual Entailment (VE) task, we utilize the pre-trained model that was obtained in the VRE task. We fine-tune this model on the SNLI-VE dataset (Xie et al., 2019) for 5 epochs and 20,500 updates. The fine-tuning process was distributed across 4 NVIDIA A40 GPU workers and took approximately 15 hours to complete. For the rest of the training process, we largely adhered to the OFA setup. For Image Captioning (IC), fine-tuning is conducted in two stages: (1) cross-entropy optimization for two epochs with a batch size of 128, learning rate of 1e-5, and label smoothing of 0.1; (2) CIDEr optimization for three additional epochs using a batch size of 64, disabling dropout and stochastic depth for stability.

# B    Qualitative Examples

We conduct an in-depth qualitative analysis to uncover potential patterns in the generated Emergent Communication (EC) text. For this analysis, we train three EC speakers over 1000 epochs, utilizing a vocabulary size of 4035 and sequence lengths of 5, 15, and 25. The speakers are trained using COCO features from the EC game introduced by (Yao et al., 2022). To generate EC text, we pass the first 1000 unique images from the refCOCO training dataset through each speaker. We track the positions of EC text n-grams within the generated sequences. If more than 5 images produce the same n-gram in the same position of the text sequence, we group those images for manual observation. Specifically, we examine bigrams, trigrams, and 4-grams for EC sequences of length 5, 15, and 25, respectively.

Figure 7 highlights groupings in EC sequences of length 5. For example, token 2430 is strongly associated with broccoli, while token 222 is frequently utilized to describe food as a broader category earlier in the sequence. In Figure 8, which presents examples with a sequence length of 15, we observe that token 3293 exhibits a strong visual grounding to zebras, suggesting consistent semantic alignment. Figure 9 further illustrates how the same tokens, when placed in different positions, convey similar yet more refined meanings. For instance, token 309 corresponds to vehicles, but its count and position within the sequence determine whether it refers to a truck or a motorbike. This behavior implies that EC tokens may encode coarse-grained semantics individually while achieving finer-grained distinctions through compositional patterns, akin to polysemy and contextual disambiguation in natural language.

Furthermore, we observe that token 3355 appears across diverse semantic contexts, hinting at its potential structural or delimitative role within the EC syntax, perhaps analogous to punctuation or function words. This pattern aligns with the hypothesis that certain tokens may be repurposed as syntactic scaffolds to organize more semantically meaningful tokens within the EC grammar.

Additionally, Figure 10 illustrates how n-gram patterns contribute to broader sequence-level structure. For instance, the bigram (1599, 1599) appears consistently across multiple sequences, sometimes at position 0 and other times at position 1. This observation suggests that its semantic contribution is preserved regardless

---

[2]`https://github.com/ysymyth/ec-nl/tree/master/ec-game`
[3]`https://github.com/OFA-Sys/OFA`

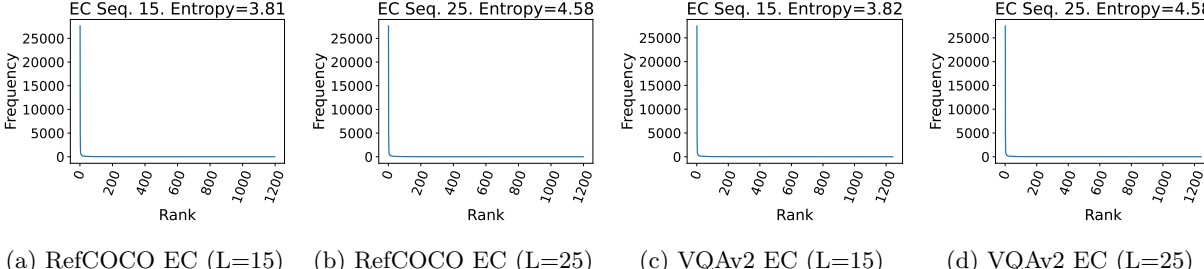

(a) RefCOCO EC (L=15)   (b) RefCOCO EC (L=25)   (c) VQAv2 EC (L=15)   (d) VQAv2 EC (L=25)

Figure 6: **Unigram distributions for EC tokens at longer sequence lengths (L=15, 25).** Compared to L=5, longer EC sequences exhibit higher entropy. EC communication remains compact and token-efficient, indicating that it is optimized for information transfer rather than expressiveness.

of its exact position. This invariance hints that n-gram identity, rather than absolute position, may carry semantic weight, or that a higher-level structural framework governs EC sequence composition.

Interestingly, some images appear in both bigram and trigram configurations of token 1599, indicating a repetition-based mechanism for refining semantic precision, *e.g.*, a trigram (1599, 1599, 1599) could emphasize a salient visual feature more strongly than a single token, akin to prosodic stress or syntactic repetition in human language. This repetition may reflect a form of emergent hierarchical encoding. Much like fixed phrases in natural language, EC subsequences may preserve meaning even when shifted in position, pointing to latent compositionality in the learned token space. Results suggest that the emergent communication protocol not only encodes discrete semantics but may also develop primitive mechanisms for emphasis, structure, or saliency highlighting. Remarkably, these patterns emerge without any predefined linguistic structure or grammar, indicating that EC self-organizes to support both grounding and discriminative specificity.

## C  Additional Ablation Results on Unigram Distribution

In Figure 6, EC unigram distributions with longer sequence lengths (L=15 and L=25) in both RefCOCO and VQAv2 reveal a clear trend toward lower entropy and flatter distributions. Entropy decreases from 4.35 (RefCOCO, L=15) to 3.98 (L=25) and from 4.40 (VQAv2, L=15) to 4.01 (L=25), indicating that usage remains concentrated among a core subset of tokens. While the vocabulary size increases, token frequency becomes more structured and repetitive, suggesting the EC relies more heavily on compositional patterns rather than expanding lexical diversity. This supports the hypothesis that EC develops structured but efficient communication protocols, optimized for information transfer rather than expressiveness.

## D  Limitations

While our work provides strong empirical evidence for the benefits of EC pre-training across several core vision-language tasks, it does not exhaustively evaluate the generalizability of EC tokens across the full spectrum of multimodal tasks. Our findings revealed significant improvements in learning and task generalization, but a comprehensive evaluation across a wider range of tasks remains an important direction for future work. A central limitation of our approach lies in its reliance on EC sequences derived from referential games grounded in literal image semantics. While effective for grounding and localization tasks, these tokens may lack the compositional structure or pragmatic nuance required for more complex settings, such as abstract reasoning, multi-turn dialogue, or instruction following. As such, substituting natural language with EC tokens is not always straightforward and may depend on task-specific demands. Moreover, while EC tokens are generated without human-annotated text and thus avoid direct exposure to textual biases (*e.g.*, gendered language or cultural metaphors), they are still grounded in visual data and may reflect biases inherent in the image distribution. For example, dataset-level imbalances in object representation, demographics, or co-occurrence patterns may shape the semantics of emergent tokens in unintended ways. We believe this highlights an important future direction: evaluating and mitigating bias in synthetic emergent communication protocols. Such future work could also serve as a testbed for studying these issues in a

controlled setting, and insights gained in EC may inform the design of future multi-agent LLM systems that rely on emergent, protocol-driven communication. Finally, our findings suggest that EC pretraining is especially promising in data-constrained settings, and we envision its broader utility when combined with structured natural language supervision. Future work should explore hybrid models that flexibly integrate EC tokens and language-based representations to benefit from the strengths of both paradigms.

## E   Broader Impact

Our work has several potential broader impacts. Firstly, the possibility of integrating emergent language pre-training from Emergent Communication (EC) into Vision Language Models (VLMs) paves the way for the development of more robust and generalizable vision-language learning methods. This could have a positive impact on various applications, such as image-text retrieval, visual search, visual question answering, and image captioning, in addition to important implications for enabling VLMs to perform effectively in real-world settings where representative data is limited, thus enhancing their practical utility.

Secondly, our research contributes to advancing communication between humans and machines. By investigating how agents learn to communicate in EC games and establishing connections with vision language systems, we gain deeper insights into the cognitive and computational mechanisms that underlie effective communication. This understanding can fuel the development of more efficient and intuitive communication systems, benefiting both humans and machines in various domains. Investigating how agents might develop compositional, grounded communication from scratch could inform the design of future human-AI interfaces that are more adaptive, interpretable, or robust in dynamic environments, such as assistive robotics or collaborative decision-making.

Third, our study lays foundational groundwork for a new pretraining paradigm, one that complements natural language supervision with self-discovered symbolic communication. By doing so, it contributes to broader efforts in artificial intelligence to develop agents that reason over multiple modalities using internal, compact representations that do not depend solely on human-curated semantics. This, in turn, contributes to the evolution of AI technologies that better understand and interact with the world around them.

We do not anticipate significant negative impacts associated with this work. However, we acknowledge that emergent communication tokens, though not authored by humans, are ultimately learned from data-driven interactions and shaped by the distribution of visual inputs, optimization objectives, and task definitions. As such, they may still encode or amplify underlying biases present in the training data. For example, if image datasets exhibit imbalance or stereotype-reinforcing correlations, EC tokens might reflect these patterns in unexpected ways. Moreover, EC tokens lack the interpretability of natural language, which may complicate transparency or accountability in downstream applications. As EC-based representations are further explored and integrated into vision-language systems, it will be crucial to develop auditing tools and safeguards to ensure their robustness, fairness, and alignment with human values. Overall, our findings highlight EC not as a substitute for natural language, but as a complementary, scalable signal that can support vision-language learning, particularly in domains where natural language is limited or unavailable.

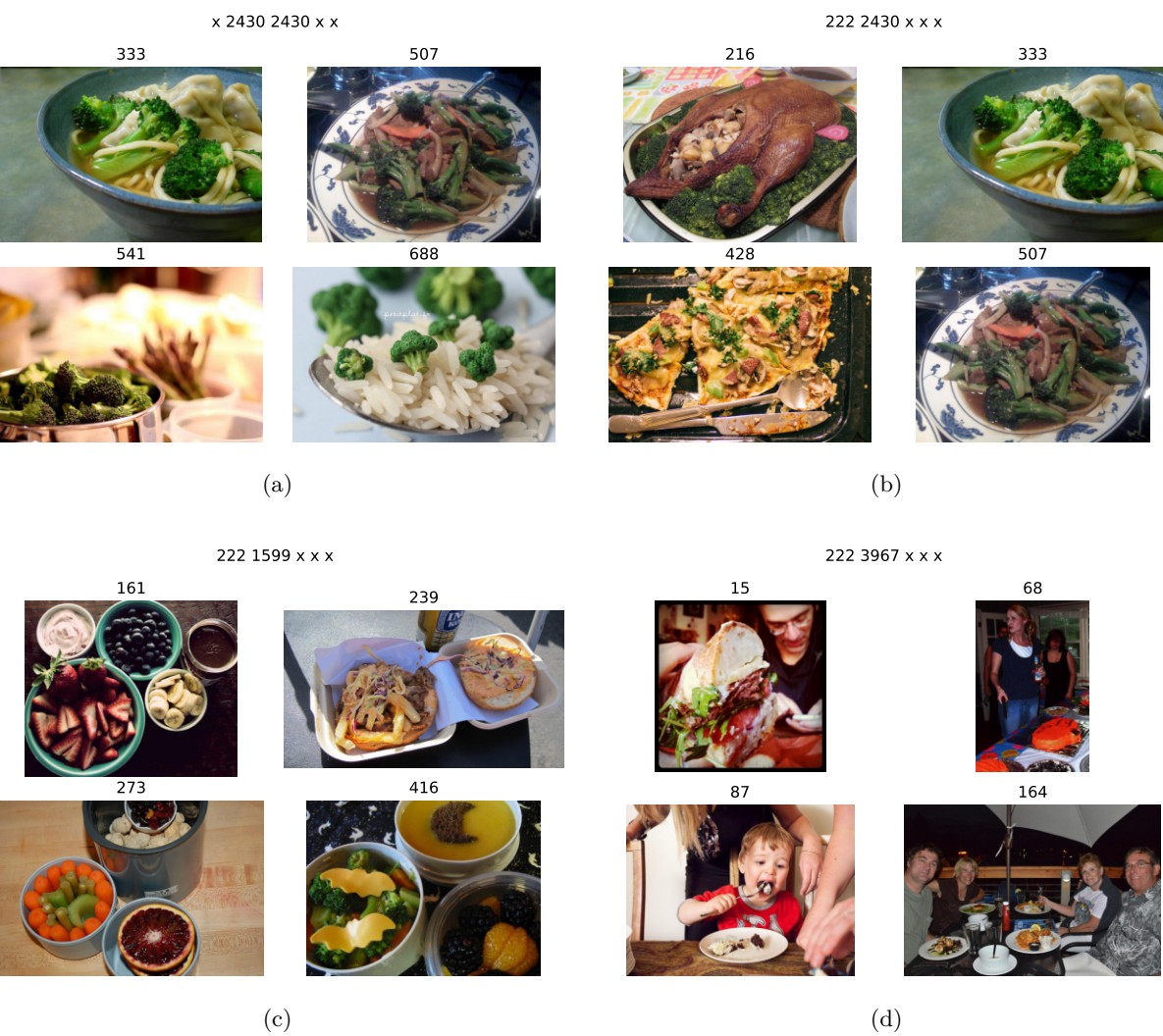

Figure 7: **Examples of EC Sequences Exhibiting Semantic Clustering.** (a) The repeated occurrence of token 2430 is consistently associated with images containing broccoli. (b) Token 222 functions as a higher-level food category marker. (c) Varying the tokens that follow 222 refines the type of food being described, suggesting contextual disambiguation. (d) The bigram 222 3967 remains food-associated but often appears in scenes involving people interacting with food, such as eating or holding it, indicating compositional encoding of both object and action.

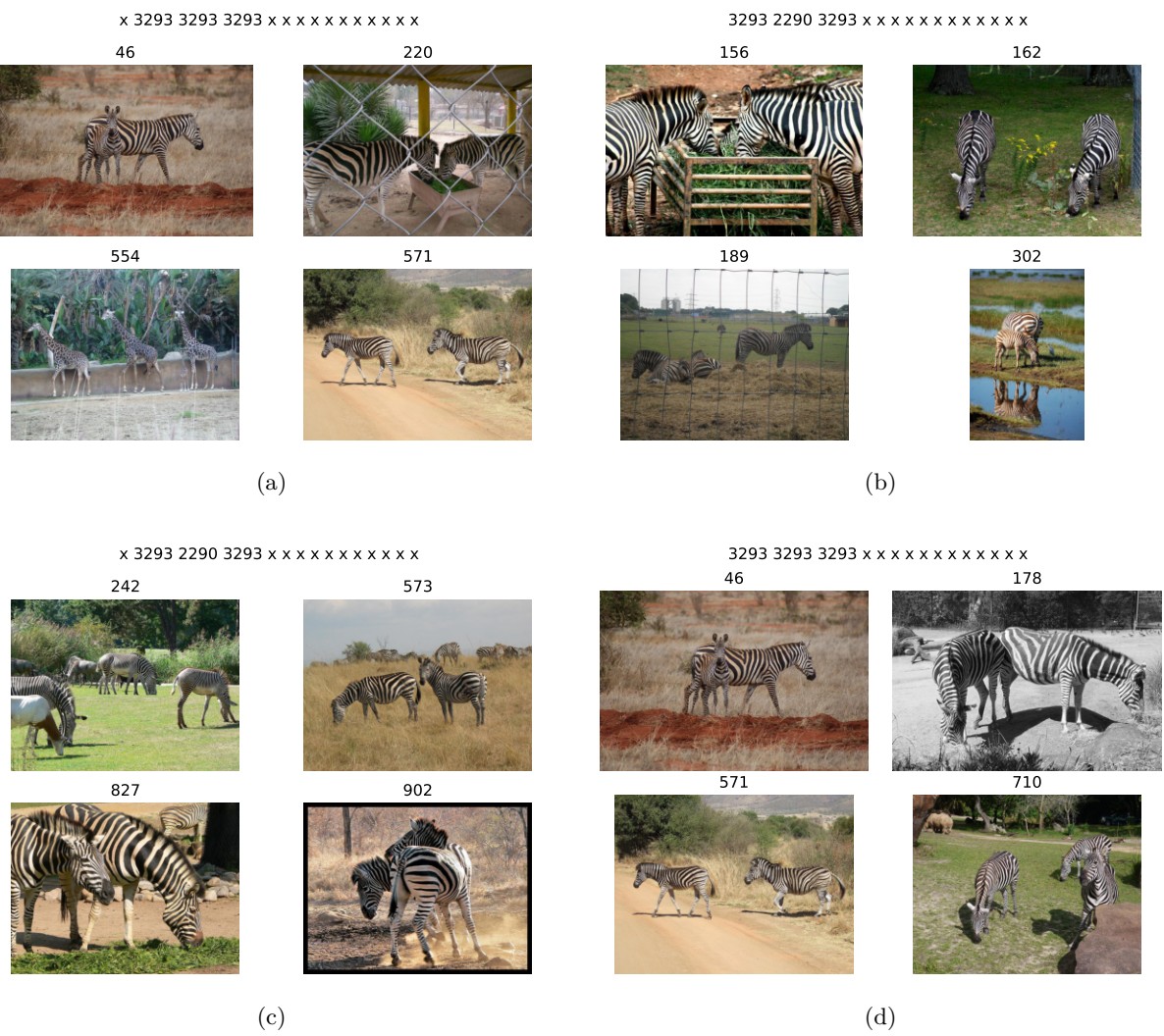

Figure 8: **EC Sequences Reveal Visual Grounding, Compositionality, and Latent Structure.** (a) Token 3293 consistently appears in zebra-related images, demonstrating strong visual grounding. (b) A variation in the third token of the trigram suggests fine-grained visual distinctions between zebra scenes, pointing to contextual compositionality. (c) The same trigram from (a) appears at a different sequence position, indicating positional flexibility and implying that EC meaning is carried by token patterns rather than fixed positions, a possible marker of syntactic invariance. Occasional co-occurrence with giraffes suggests token reuse across visually related concepts and hints at fuzzy semantic boundaries between visually similar classes. (d) The trigram from (b), when shifted to position 1, remains strongly correlated with zebra images, reinforcing compositional consistency and semantic robustness.

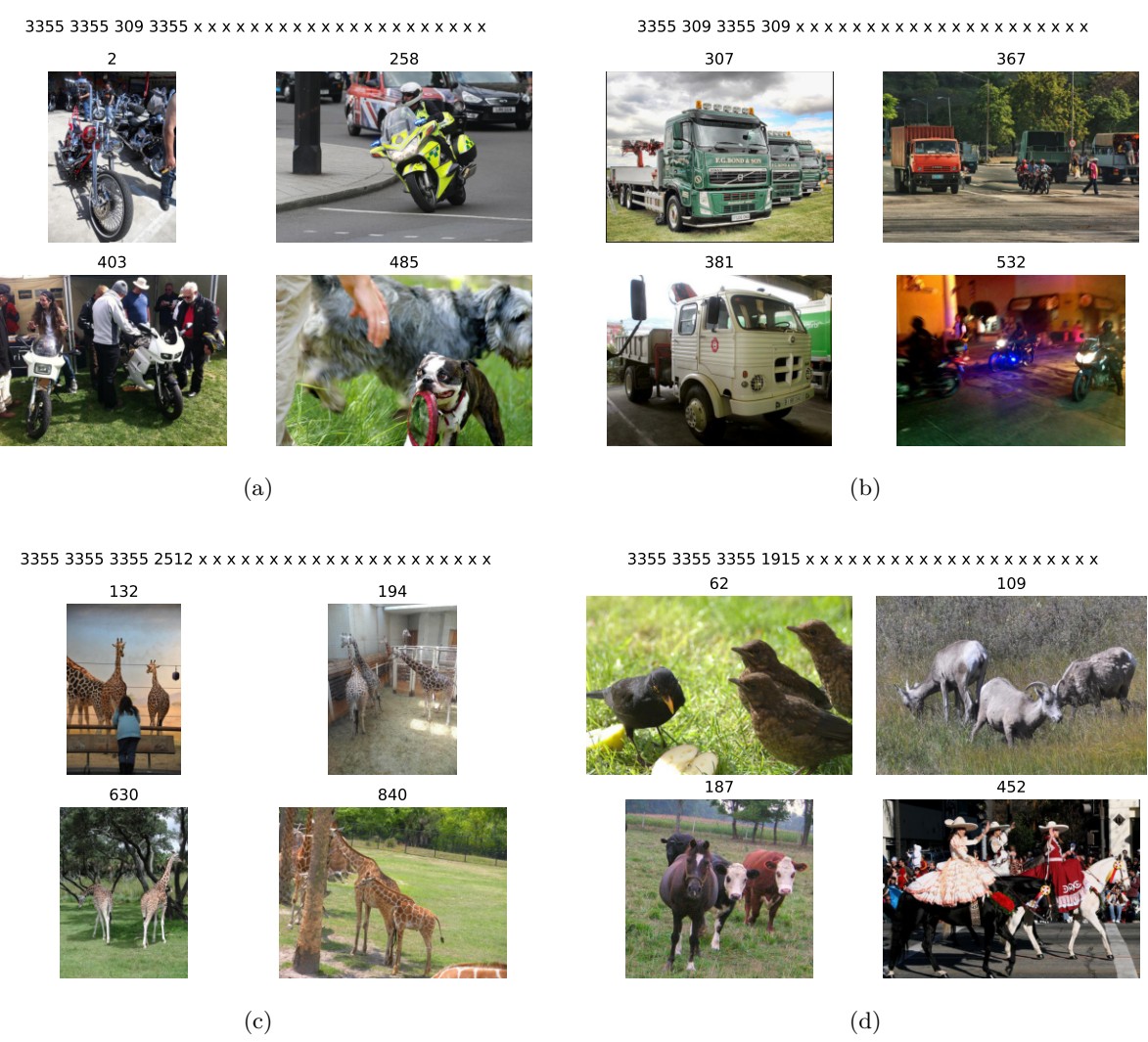

Figure 9: **EC Sequences Exhibit Semantic Specificity and Structural Roles.** (a) Token 309 is associated with vehicles, and its position appears to modulate meaning, *e.g.*, at position 2, it tends to refer to motorbikes. (b) The same token in different positions corresponds to trucks, suggesting context-dependent semantic refinement. (c) Token 2512 may denote giraffes, while (d) token 1915 appears to generalize to a broader animal category. Notably, token 3355 occurs across all examples, suggesting a structural or functional role, potentially indicating count, emphasis, or grouping within the sequence.

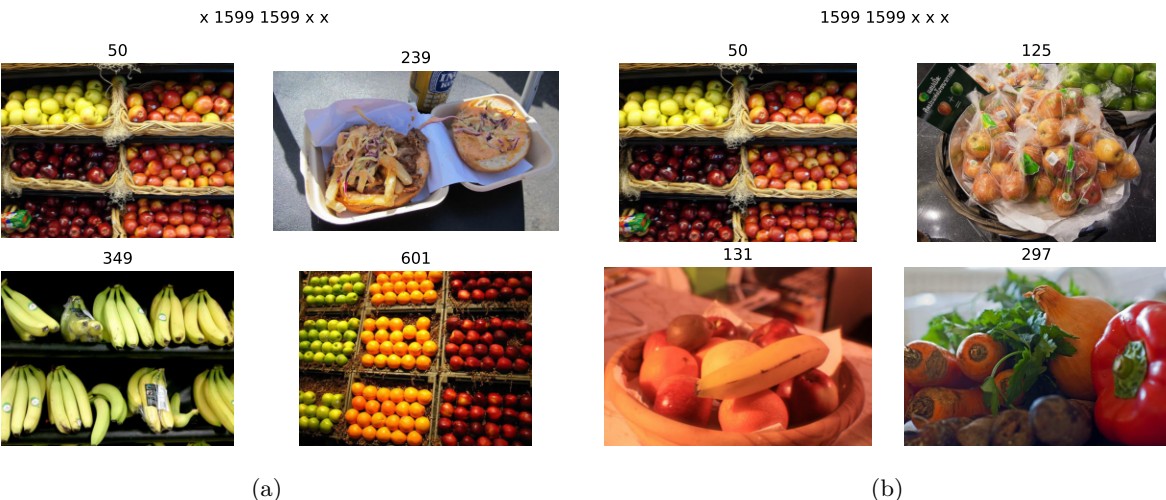

Figure 10: **Semantic Consistency and Repetition Effects in EC Sequences.** Images are grouped based on the occurrence of the bigram (1599, 1599), which consistently appears in EC sequences describing scenes with vibrant red, yellow, and green produce. (a) When this bigram appears in position 0, it strongly correlates with fruits and vegetables featuring these colors. (b) When shifted to position 1, the bigram still retrieves visually related scenes, though with slightly more variation. These patterns suggest that certain EC n-grams carry consistent semantics across positions and that repetition may emphasize salient visual features akin to prosodic stress or syntactic repetition in natural language.

