# OpenReview forum: "Emergent Corpus Pre-training Benefits Vision Language Models"
_TMLR — Accepted by TMLR_

### Review · Reviewer_SkkB · 2025-04-17

**Summary Of Contributions:**

The paper introduces Emergent Communication (EC) as a novel mechanism to improve sample efficiency in Vision-Language Pre-trained Models (VL-PTMs). The authors propose a framework where EC tokens are generated through a referential game between two artificial agents, and these tokens are used as pretraining signals for vision-language models. The major contributions of the paper are as follows: Introduction of EC in Vision-Language Pretraining: The authors leverage EC tokens generated through a referential game between agents as an effective pretraining signal for downstream vision-language tasks, including Visual Referring Expression (VRE), Visual Entailment (VE), and Visual Question Answering (VQA). Empirical Validation: The paper demonstrates the effectiveness of EC pretraining through a series of experiments, including substantial gains in performance on diverse tasks: RE accuracy improved by 108.6%. VE accuracy improved by 69.6%. LLaVA-1.5-EC, a model trained entirely on EC tokens, outperforms strong baselines (e.g., BLIP-2) on multiple benchmarks, including VizWiz, GQA, and VQAv2. Introduction of LLaVA-1.5-EC: A LLaVA variant that is pretrained solely on EC tokens, highlighting the potential of EC in low-resource and text-scarce settings. Theoretical Insights: The paper explores the compositional structure of EC tokens, their capacity to encode grounded semantic information, and their ability to generalize across different tasks. These contributions propose EC as a scalable, annotation-free signal that can improve vision-language modeling, especially in settings with limited data.

**Audience:**

Yes

**Claims And Evidence:**

Yes

**Requested Changes:**

1. Think carefully about weakness.
2. Table 3 should show good results in bold.
3. The words in the pictures in Figure 5 and Figure 6 are too small.
4. Code links can be placed in the body.

**Strengths And Weaknesses:**

Strengths:

Innovative Pretraining Mechanism: The use of EC tokens for pretraining vision-language models is a novel approach that significantly enhances the sample efficiency of the models.

Strong Empirical Results: The authors provide compelling experimental evidence showing that EC pretraining leads to substantial improvements across several vision-language tasks, especially in low-resource settings.

Scalability and Generalization: The paper convincingly demonstrates that models pretrained with EC tokens, such as LLaVA-1.5-EC, can achieve competitive performance on instruction-following tasks, despite being trained on synthetic data without human annotations.

Clear and Structured Approach: The framework for EC pretraining and the experimental setup are well-articulated, making it easy to follow and replicate the experiments.

Weaknesses:

Limited Comparison with Other EC-based Models: While the paper compares EC-based pretraining with natural language pretraining, it would benefit from a more detailed comparison to other emergent communication-based methods or architectures.

Lack of Comprehensive Ablation Studies: Although the paper discusses the importance of the EC structure, it would be valuable to see more detailed ablation studies exploring the impact of varying EC token lengths, agent interactions, or other hyperparameters.

Ethical Considerations: While the paper addresses the efficacy of EC pretraining, there is limited discussion on the ethical implications of using synthetic communication protocols and the potential for unintended biases in EC-generated tokens.

---

> ### Author Response · Authors · 2025-05-22
>
> Thank you for your review. We appreciate your positive feedback on the innovative pretraining mechanism introduced in our work, the comments on the strong empirical results, as well as the clarity of our approach. We address the requested changes below:
>
> **Lack of Comprehensive Ablation Studies:** We have added three new ablation studies in addition to the original two presented in the main paper. These experiments investigate how key design parameters—token sequence length (L) and vocabulary size (V)—affect the structure and effectiveness of EC tokens. Specifically, we vary $L \in {5, 15, 25}$ and train speaker-listener agents to generate corresponding EC corpora, which are then used to pretrain LLaVA-EC models. We evaluate each model on MMBench (EN) and MMBench (CN). Similarly, we fix $L=15$ and vary the vocabulary size $V \in {1000, 4035, 10000}$ in the Speaker model to study the effect of expressive capacity. Finally, we analyze channel utilization, measuring token entropy across configurations to assess whether the emergent communication protocols exhibit meaningful symbol reuse or degenerate behavior. Results reveal a clear channel capacity–generalization tradeoff, suggesting that constrained protocols may induce more abstract, compositional representations that are better suited for transfer. These findings corroborate prior EC work showing that, in the absence of regularization, agents often exploit available capacity in ways that are effective for local coordination but misaligned with compositional or transferable representations, raising new questions about how to regularize communication protocols for optimal vision-language downstream utility. We detail ablation results in our revised manuscript under subsection “Channel Capacity–Generalization Tradeoff in EC Token Design.”, Figure 5 and Table 7.
>
> **Comparison to Other EC-based Methods:** We thank the reviewer for raising this point. Most existing emergent communication (EC) works focus on analyzing protocol emergence in symbolic or synthetic environments, such as referential games over color-shape grids or navigation tasks in limited domains. These works typically use EC to study linguistic emergence and compositionality, rather than as a tool for pretraining scalable multimodal models. To our knowledge, our work is the first to explore the use of EC tokens as a pretraining signal for vision-language models targeting multi-modal reasoning tasks. Since prior EC studies do not address this setting, nor are they evaluated within the VLM framework, a direct empirical comparison is not feasible. Our comparisons against natural language-pretrained VLMs (OFA, LLaVA-1.5) are precisely chosen to demonstrate the efficacy and transferability of EC as a substitute or complement to human supervision in this novel application.
>
> **Ethical Considerations:**  Thank you for raising this important point. Synthetic communication is indeed a novel paradigm in vision-language training, and we appreciate the opportunity to reflect on its ethical dimensions. EC tokens represent visual semantics directly, without conditioning on human-annotated text, offering a potential advantage in reducing textual biases such as gendered language or culturally specific metaphors. However, we acknowledge that EC-generated tokens are still grounded in visual data and may reflect dataset-level biases present in the images themselves. We have refined our “Limitation” section to have a thorough discussion of this.
>
> **Requested Changes:** All minor requested changes have been effected in the paper. Table 3 now shows results in a better format so that good results can be easily identified. Figures 5 and 6 (corresponding to Figures 4 and 6 after our new revisions) are now fixed, and code links will be included in the camera-ready version.

---

> > ### Comment · Reviewer_SkkB · 2025-05-24
> >
> > “Limited comparison with other EC‐based models.” The paper should include a more detailed empirical comparison against existing EC frameworks or architectures.
> >
> > Insufficient justification:
> > • No concrete references are cited to prove irrelevance or infeasibility.
> > • No alternative sanity tests (e.g., adapting a classic EC referential-game protocol to a small VLM) are provided.

---

> > > ### Author Response · Authors · 2025-05-25
> > > **comment**
> > >
> > > Thank you for the follow-up. We reiterate that the **referential game protocol we adopt is standard in the emergent communication literature** [1,2]. Our contribution is not in proposing a new game, but in demonstrating that this classical EC setup can be repurposed to generate EC tokens that effectively supports VLM pretraining, including OFA (**a small 180M parameter VLM already provided in our paper**) and LLaVA-1.5 (a state-of-the-art instruction-following 7B model). If the reviewer has **concrete guidance** on a specific EC-based model in mind that could serve as a meaningful point of comparison, we would be happy to consider it.
> > >
> > > Finally, in response to the request regarding code availability, we have now released a temporary anonymous code repository: [LINK](https://anonymous.4open.science/r/EC-VLM-D07E/README.md)
> > >
> > > [1] Lazaridou, Angeliki, Karl Moritz Hermann, Karl Tuyls, and Stephen Clark. “Emergence of Linguistic Communication from Referential Games with Symbolic and Pixel Input.” In International Conference on Learning Representations (ICLR) 2018
> > >
> > > [2] Yao, Shunyu, Mo Yu, Yang Zhang, Karthik R. Narasimhan, Joshua B. Tenenbaum, and Chuang Gan. “Linking Emergent and Natural Languages via Corpus Transfer.” In International Conference on Learning Representations (ICLR) 2022

---

> > > > ### Comment · Reviewer_SkkB · 2025-05-25
> > > >
> > > > OK, I have no more questions.

---

> > ### Comment · Reviewer_SkkB · 2025-05-24
> >
> > A temporary anonymous repository (GitHub/Zenodo) should be shared immediately.

---

### Review · Reviewer_o3BJ · 2025-04-28

**Summary Of Contributions:**

This work studies improve downstream tasks of large language models in low-resource settings with EC. The proposed method adopts RL method to inject visual prior knowledge from the vision model into the large language model through discrete "pseudo-language" tokens. This provides better visual understanding and initialization. The proposed method improves the fine-tuning performance on several downstream tasks, working as an alternative for the limited natural language annotations scenarios.

**Audience:**

Yes

**Broader Impact Concerns:**

The reviewer does not have major ethical concerns about this work. The proposed method focuses on reducing reliance on large-scale human annotations, which could potentially lower data collection costs and promote accessibility in low-resource settings for the MLLM. The use of artificial agents for token generation does not raise obvious ethical risks.

**Claims And Evidence:**

Yes

**Requested Changes:**

* It seems that the proposed method adopts EC tokens in pretraining but still needs fine-tuning on downstream tasks, which poses a gap that the EC pretraining could harm the original multimodal alignment ability or other abilities like reasoning of MLLMs.  How to solve potential limitations remains unclear.

* The generation of EC tokens requires two agents with repeated interaction and shared supervision. The construction cost is important but remains unclear.

* It is not clear how the proposed method can be deployed in open scenarios mentioned in conclusion. Besides, the performance gain of the proposed method in high-resource settings is unknown.

* It seems that there lacks comparisons with other weakly supervised strategies. It is of the interest to see how the proposed method performs against contrastive learning under the same low-resource setting.

**Strengths And Weaknesses:**

* The reviewer acknowledges that the proposed method offers a way to reduce the need for large-scale image-text annotations in low-resource settings.

* The pretrained model outperforms some strong supervised methods on certain tasks.

* The proposed method can serve as a complementary pretraining strategy.

---

> ### Author Response · Authors · 2025-05-22
>
> Thank you for the encouraging review recognizing our paper's strong performance and potential impact. We address comments below:
>
> **Potential Limitations of EC Tokens in Pretraining and Finetuning on Downstream Tasks:** Thank you for this insightful comment. Fine-tuning on downstream tasks is a standard practice to bridge the gap between pretraining and task-specific adaptation, as seen in models like CLIP, BLIP, and InstructBLIP. Similarly, in our case, EC tokens serve as compressed representations of visual semantics that complement, rather than conflict with, multimodal alignment objectives. This mirrors prior approaches that leverage vision-centric or contrastive pretraining (e.g., BLIP, Flamingo) before incorporating language supervision. Importantly, we observe that models pretrained with EC tokens transfer well across tasks, for example, a model trained on VRE generalizes effectively to visual entailment without task-specific modifications, suggesting that EC tokens capture transferable structure. Subsequent fine-tuning with natural language helps recover broader alignment while preserving the inductive benefits of EC-based grounding. We agree that further analysis is warranted to better characterize these trade-offs, and we view hybrid EC+NL pretraining strategies as a promising future direction to combine the structure-inducing properties of EC with the expressivity of language.
>
> **Construction Cost of EC Tokens:** Thank you for raising this point. EC token generation occurs entirely offline during pretraining and involves a one-time small computational cost. In contrast, collecting human-annotated captions incurs ongoing cost and effort as datasets scale. The speaker-listener interaction is fully automated and scales linearly with the number of images. In our case, generating the full EC corpus required approximately 2 hours on one NVIDIA A40S GPU. Assembling a comparable-scale image-text corpus would typically involve days to months of manual annotation. We believe this highlights the efficiency and scalability of EC-based pretraining.
>
> **Open Scenarios and High-Resource Settings:** Thank you for highlighting these important points. Regarding deployment in open scenarios, we clarify that EC token generation is fully automated, annotation-free, and scalable, making it well-suited for continual or self-supervised learning in open-world environments where human supervision is limited or unavailable. Our conclusion speculates on future directions, such as evolving EC into an expressive representational language or embedding it within agent ecosystems. While exploratory, these ideas are grounded in the demonstrated efficiency and transferability of EC tokens. As for high-resource settings, our current focus is on low-resource and weakly supervised regimes, where EC provides the most immediate practical benefit. EC pretraining could serve as an additional modality-specific grounding signal that improves sample efficiency and generalization, particularly before large-scale language tuning. For example, given that EC tokens are language-agnostic, whereas in high-resource pipelines natural language annotations are often biased toward English or specific domains, EC can improve compositional grounding, enable curriculum-style pretraining, and support multilingual or cross-domain alignment. While this paper focuses on the low-resource case to emphasize the standalone value of EC, integrating EC pretraining into high-resource pipelines is a promising direction for future work.
>
> **Comparison with other Weakly Supervised Strategies:** We thank the reviewer for this valuable suggestion. In response, we include a new comparison between our proposed EC-based pretraining method (LLaVA-EC) and a contrastive learning baseline modeled after CLIP. Specifically, we compare LLaVA-EC, in which image features are aligned to the EC semantic space before being passed to the language model, against a new baseline we term LLaVA-1.5-CLIP-7B, where CLIP image embeddings are projected directly into the LLM input space. This comparison allows us to isolate the benefits of EC token grounding relative to standard contrastive vision encoders. Results are provided in the updated manuscript Table 5 and Figure 3.

---

### Review · Reviewer_BYq9 · 2025-05-07

**Summary Of Contributions:**

This paper presents using the Emergent Corpus from a referential game as a substitute for image-text pairs for pre-training vision-language models. The main contribution is that EC tokens enable an effective alternative when high-quality image-text pairs are limited.

The experiment results demonstrate consistent improvements on a variety of tasks, including visual referring expression, visual question answering, visual entailment, image captioning, and instruction-following tasks, despite still seeing a relatively large gap with using natural language. Moreover, for visual entailment, a promising scalability of pre-training with EC tokens is shown, and the performance converges to that of using natural language.

Overall, the proposed use of emergent communication for vision-language pre-training presents an interesting direction for synthetic data generation. Although the performance still lags when using natural language, this paper serves as a proof of concept.

**Audience:**

Yes

**Broader Impact Concerns:**

The authors include a comprehensive section on the broader impact concerns, and I have no other concerns.

**Claims And Evidence:**

Yes

**Requested Changes:**

**Figures:**
1. Figure 2 does not provide any useful information, as the emergent expressions are not natural language. Including them here is confusing.
2. The plots in Figure 5 are hard to distinguish. Using a table to show the entropies is better.

**Experiments:**
1. See Weaknesses Part for the experiments recommended to add in the paper.

**Strengths And Weaknesses:**

**Strengths:**
1. The training of the speaker and listener agents is self-supervised, which makes the approach highly scalable.
2. Using EC tokens for pre-training circumvents the need for manual curation of image-text pairs, equivalent to generating synthetic data with image inputs only.
3. The experimental results show some potentials of EC tokens in pre-training vision-language models.

**Weaknesses:**
1. Including an experiment on image retrieval using learned EC tokens can be beneficial to demonstrating the effectiveness of EC tokens. A test on retrieval is necessary as it directly relates to the referential game used to train agents for EC token generation.
2. When EC pretraining is used during the feature alignment stage of LLaVA-1.5, the CLIP image features are aligned to EC tokens rather than to natural language. To demonstrate the effectiveness of EC tokens, an ablation study that directly fine-tunes LLaVA-1.5 for visual instruction tuning, bypassing the feature alignment stage, is necessary. This comparison helps validate that EC tokens provide a more effective representation than raw CLIP features for instruction tuning. Ideally, the results should show that EC tokens are more effective than raw CLIP features despite lagging natural language.
3. Some commonly used benchmarks are not included in instruction tuning tasks, such as MMMU, SQA, SEED, etc. Please refer to the original LLaVA paper and include the missing benchmarks for a more comprehensive comparison.

**Questions:**
1. For training the speaker and listener agents, the training objective is purely based on selecting the true image from a set. How do the authors recognise their learned EC tokens as an emergent language rather than some kind of discrete visual tokens similar to VQ-VAE and VQGAN? (From the second ablation on the structure of EC tokens, it seems that random order leads to a rather small decrease in performance.)
2. The images in the image set for selection when training the agents are randomly selected. Have the authors tried hard sample mining by selecting images that are more semantically similar with the query image (by CLIP features, etc.), therefore potentially leading to more fine-grained details in EC tokens?

---

> ### Author Response · Authors · 2025-05-22
>
> **Image Retrieval:** We appreciate the reviewer’s suggestion. While we agree that retrieval is a valuable test of representation quality, we note that the referential game framework already serves as a retrieval task in a more structured and goal-driven form. In this setup, the Listener agent must select the correct image from a set of distractors based solely on the EC token sequence produced by the Speaker. The referential game success rate, reported in Table 7 of our revised manuscript, quantifies the percentage of successful selections and reflects how effectively EC tokens support fine-grained visual discrimination. Importantly, because we report success rates across different EC token lengths and vocabulary sizes, this measure also provides insight into the channel capacity vs. generalization tradeoff. Thus, the referential game not only serves as a retrieval proxy but also offers a window into how emergent communication efficiency relates to transferability.
>
> **CLIP:** We thank the reviewer for suggesting this ablation. To directly assess the impact of EC-based feature alignment, we implemented a LLaVA-1.5-CLIP-7B baseline, which bypasses the EC token stage entirely. In this variant, we feed pretrained CLIP image embeddings directly into the LLM via a randomly initialized projection layer, trained end-to-end during instruction tuning. This setup mirrors standard LLaVA initialization practice when no intermediate alignment tokens (e.g., EC or natural language) are used. All other components, i.e., model architecture, training configuration, and datasets, were kept identical to our LLaVA-1.5-EC pipeline to ensure a controlled comparison. We report the results in Table 5 and Figure 3. Across multiple benchmarks, LLaVA-1.5-EC consistently outperforms this CLIP baseline, demonstrating that EC-based pretraining provides a more semantically structured and transferable representation than direct projection from CLIP features alone.
>
> **LLaVA Benchmarks:** In our initial version, we incorporated seven (7) benchmarks from the original LLaVA paper, including VQAv2, GQA, VizWiz, SciQA, MMBench (EN), MMBench (CN), and TextVQA. Also, we already have the evaluation on the SQA benchmark, referred to as SciQA in our paper. We have now included additional evaluations on POPE, MME, and MM-VET, presented in Table 5 in our revised manuscript and discussed in subsection 5.5 Instruction-Following Tasks.
>
> **Requested Changes:** Thank you for your requested changes and suggestions. We have removed Figure 2. We have also revised distribution plots for clarity. Our goal was to visualize the Zipfian-like unigram distributions of EC tokens, which exhibit long-tailed frequency patterns common in natural and emergent languages. Due to the large number of token types and the importance of illustrating frequency skew, we find that plots were more effective than tables for conveying this information.

---

> > ### Author Response · Authors · 2025-05-22
> >
> > **Q1.** We appreciate the reviewer’s thoughtful question. While our training objective is based on referential success, we argue that the resulting EC tokens exhibit properties of an emergent communication protocol, rather than acting as a static visual codebook like VQ-VAE or VQGAN. First, unlike VQ-based methods where token assignments are fixed post-quantization, our EC tokens are generated dynamically by a trained Speaker agent and are interpretable by a separately trained Listener, suggesting that these tokens acquire meaning through interaction and use, a core characteristic of emergent language.Unlike VQ-VAE or VQGAN, where tokens are optimized for reconstruction or compression, EC tokens are optimized through communication success, making them functionally grounded and dynamically shaped by listener feedback. Second, the EC tokens support transfer to unseen downstream tasks, including visual question answering, entailment, and instruction following, tasks never seen during EC generation. This transferability suggests that the tokens capture semantic regularities beyond memorization or local encoding. Regarding the random order ablation: we agree that the performance drop is modest, but it is consistent across benchmarks, indicating some level of token order dependence. While this does not yet qualify as a fully developed language, we see it as an emergent proto-language, a set of discrete, context-sensitive symbols developed under communicative pressure, rather than reconstruction.
> >
> >
> > **Q2.** Thank you for the thoughtful suggestion. In our current setup, distractor images for the referential game are selected uniformly at random from the dataset. This design choice was intentional, aimed at avoiding potential bias from pre-existing semantic embeddings that could leak structured information into the EC training process. That said, we agree that incorporating semantically similar distractors could create a more challenging referential game, thereby encouraging the emergence of more discriminative and fine-grained EC tokens. Such distractors would increase communicative pressure on the agents to disambiguate subtle visual differences, potentially enhancing both the semantic expressiveness of EC tokens and their transferability to downstream tasks requiring finer visual distinctions.
> > Interestingly, prior work supports the intuition that increased task difficulty and more confounding distractors lead to more expressive and compositional communication. Guo et al. (2023) [1] explicitly construct hard distractors by designing semantically similar rule variants, showing that communicative precision improves under such conditions. Similarly, Chaabouni et al. (2022) [2] demonstrate that increasing the number of candidate distractors increases pressure for more generalizable and differentiated protocols. Motivated by these findings, we hypothesize that applying hard sample mining, for example, by selecting distractors based on CLIP similarity, could achieve similar gains in our setting. We view this as a promising direction for future work to improve the granularity and compositionality of EC pretraining, particularly for fine-grained downstream vision-language tasks.
> >
> > [1] Chaabouni, R., Strub, F., Altché, F., Tarassov, E., Tallec, C., Davoodi, E., Mathewson, K.W., Tieleman, O., Lazaridou, A. and Piot, B., Emergent communication at scale. In ICLR 2022
> >
> >
> > [2] Guo, Y., Hao, Y., Zhang, R., Zhou, E., Du, Z., Song, X., Wen, Y., Zhao, Y., Zhou, X., Guo, J. and Yi, Q., Emergent communication for rules reasoning. In NeurIPS 2023

---

### Decision · Action_Editor_f9wP · 2025-07-02

**Recommendation:** Accept as is

**Additional Comments:**

Please consider adding more discussions on the limitation of the paper.

**Audience:**

Yes

**Audience Explanation:**

The paper addresses the challenge of reducing dependence on large-scale image-text data using a creative and scalable approach. Its integration of EC into modern VLMs is novel and relevant, and it opens up new directions in self-supervised and weakly supervised multimodal learning.

**Claims And Evidence:**

Yes

**Claims Explanation:**

The paper introduces a novel pretraining method using Emergent Communication (EC) tokens generated via referential games, and demonstrates consistent gains across multiple vision-language tasks, especially in low-resource settings. Results with LLaVA-1.5-EC are compelling, and comparisons to contrastive baselines help validate the method. While concerns remain about potential trade-offs (e.g., alignment or reasoning capacity), these are acknowledged and partially addressed with ablations and discussion.